# The geological structures of the Pyrenees and its peripheral basins examined through EMAG2v2 magnetic data.

África Gamisel-Muzás[1,2], Ruth Soto[2], Conxi Ayala[3], Tania Mochales[2], Félix M. Rubio[2], Pilar Clariana[2], Carmen Rey-Moral[2], Juliana Martín-León[2].

[1] Instituto Andaluz de Ciencias de la Tierra, IACT-CSIC, 18100 Armilla, Granada. a.gamisel@csic.es.
[2] Instituto Geológico y Minero de España (IGME), CSIC, 28003, 28760, 50059. r.soto@igme.es, t.mochales@igme.es, fm.rubio@igme.es, p.clariana@igme.es, c.rey@igme.es, j.martin@igme.es.
[3] Geosciences Barcelona (GEO3BCN), CSIC, Lluís Solé i Sabarís s/n, 08028 Barcelona. cayala@geo3bcn.csic.es

*Correspondence to:* A. Gamisel-Muzás (a.gamisel@csic.es)

**Abstract.** The major goal of this work is to provide an insight into the structural and geological anatomy of the Pyrenees based on the magnetic anomalies and inferred lineaments from the Earth Magnetic Anomaly Grid 2-arc-minute resolution (EMAG2v2) magnetic data. We focused on providing qualitative and semi-quantitative evidence on the magnetic signature of the Pyrenees Mountain Range domains and structures. The integration of reduced to the pole and processed maps, as well as the Bouguer anomaly map with geological data, has proved to be significantly useful in order to shed light on the main anomaly sources. Considering their magnetic response and texture, several anomalies can be linked to buried geological bodies or changes in the magnetic character of the basement. We have estimated their source bodies depth through Euler and power spectrum calculations.

We have identified and confirmed eight magnetic zones with different features and interpreted them in terms of the geological and structural setting of the area. One on the Aquitaine Basin with the highest magnetic response in the area, linked to the Sub-Pyrenean Thrust; three along the North Pyrenean Zone linked to mantle materials; one to the east of the Axial Zone representing the boundary between the Pyrenean Mountain Range and the Mediterranean Sea; two more located in the Catalan Coastal Range related to the volcanic fields in the area; one in the Gulf of Lyon caused by a thicker crustal block. The result is an overall interpretation of the Pyrenees main magnetic domains.

## 1. Introduction

The characterization of the Earth's crustal structure and composition through geophysical data has remained a significant challenge in geoscience over the last century. Among various geophysical exploration methods, aeromagnetic surveying has emerged during the last century, initially performed during the World War II (Nabighian et al., 2005). Magnetic surveying has proven to be highly effective to investigate the Earth's crust at different scales, from mapping basement structures to constraining the extent of saline diapirs (Kimbell et al., 2002; Ayala et al., 2000; Bertrand et al., 2020; Al-Bahadily et al., 2024). The aeromagnetic data surveying allows to cover large areas in a short period of time without being invasive (e.g. Vine and Matthews, 1963; Vine, 1966; Gibson and Millegan, 1998; Langel and Hinze, 1998; Golynsky, 2002; Purucker and Whaler, 2007; Maus et al., 2009). The processing of the acquired data allows generating magnetic anomaly maps that provide valuable information concerning both continental terrains and Earth's oceanic floors (e.g. Müller et al., 2008). In particular, large-scale magnetic anomaly maps are of great interest for analyzing major lineaments and structural trends (e.g. Maus et al., 2009) that can be used to improve geological mapping (e. g. Ayala et al., 2000, García-Lobón et al., 2007).

Concerning the Pyrenees, numerous geological and geophysical studies have been conducted to infer their crustal structure at depth (e.g. ECORS-Pyrenees Team, 1988; Casas et al., 1997; Chevrot et al., 2014, 2018). However, their overall study based on magnetic data has received less attention (Zeyen and Banda, 1989; Zeyen et al., 1991).

In this study we utilize data from the Earth Magnetic Anomaly Grid (EMAG2v2) (Maus et al., 2009), to interpret the magnetic patterns related to the geological structure of the Pyrenees. This study highlights the effectiveness of magnetic anomaly maps in interpreting the structural characteristics of the Pyrenees and gives an overall view of its magnetic complexity.

## 2. Data and methodology.

The magnetic data used in this analysis came from EMAG2v2 (2009) version instead of the EMAG2v3 (Meyer et al., 2017) due to the lack of data of the latter in the study area. The homogenized grid of the EMAG2v2 incorporates the aeromagnetic survey data from France (Institut du Physique du Globe de Paris – Bureau de Recherches Géologiques et Minières, 1960) and Spain (Socías et al., 1991 - Instituto Geográfico Nacional, 1986-1987) as well as international satellite and ship data. It consists of a homogenized grid displaying the magnetic

anomalies at 4000 meters above the geoid with a spatial resolution of 2-arc-min (equivalent, at this latitude, to ~3700 meters) (Maus et al. 2009). The magnetic anomaly map of the studied area has been integrated into a digital database to compare it with the main Pyrenean structures, gravimetric anomalies and the main outcrops of intrusive and volcanic rocks.

To enhance the interpretation of the magnetic signal of the Pyrenees, derivative maps and estimated calculations

of the location and depth of the magnetic sources have been generated from the total magnetic field. This enables us to improve and complete the correlation between geophysical and geological information. These calculations were made through the Oasis Montaj© software by Seequent (Hinze et al., 2013) whose formulae is primarily based on the works of Blakely (1995), Thomson (1982) and Spector and Grant (1970) and further developed considering the references included in the description of each methodology.

**2.1. Reduction to pole (RTP).**

This magnetic transformation aims to eliminate the inherent asymmetry present in total magnetic field anomalies, locating the anomalies above the causative bodies. This is achieved by converting the observed anomaly to the equivalent anomaly measured at the North Magnetic Pole (Baranov 1957; Baranov and Naudy, 1964). This transformation assumes that the remnant magnetization is negligible compared to the induced magnetization. The

advantage of this transformation is that the resulting map is easier to correlate with the surface geology and other geophysical data, like gravity, which helps with the interpretation.

**2.2. Magnetic derivatives.**

The magnetic derivatives have been proved effective to locate the diverse anomaly sources since they accentuate the outline of their boundaries, and sometimes even estimate their depth, the inclination and susceptibility contrast

between rocks (e.g. Verduzco et al., 2004). In this study, we have applied these calculations on the reduced to the pole magnetic field.

The vertical derivative (VDR_Z) can highlight the shallowest anomaly sources because enhances the high wavenumber component of the spectrum and exposes positive values above the sources. The horizontal derivative (HDR_X and HDR_Y) outlines the anomaly source bodies since it produces a phase transformation and enhances

the higher frequencies (Fanton et al., 2014). The direction of the attributes to highlight plays a crucial role in the latter since it exposes the directional variation of the magnetic field with regard to the magnetic causative bodies,

providing a distinctive anomaly texture (Hayatudeen et al., 2021). The horizontal derivative in X (N90ºE) direction will enhance N-S discontinuities and the Y (N180ºE) direction will enhance E-W discontinuities. Additionally, directional derivatives in N110ºE and N200ºE would enhance the structures following the main Pyrenean directions. As a combination of both vertical and horizontal derivatives of the magnetic anomalies, the analytic signal shows symmetric maximum values above the anomaly sources (Ansari and Alamdar, 2009).

Initially described by Miller and Singh (1994) as an innovative method to distinguish the anomalies as well as their edges and peaks, the tilt derivative (TDR) combines the horizontal and vertical derivatives, later refined by Verduzco et al., (2004). Consequently, the tilt angle answers to both horizontal and vertical derivatives on magnetization, so weak magnetic bodies are weighted the same as strongly magnetized ones (Blakely et al., 2016). The vertical derivative positive values peak above the source has zero value along its edge and turns negative outside of the source. The horizontal gradient is positive along the edges and is zero above the source (Miller and Singh, 1994). The tilt angle derivative is zero on the edge of the anomalies. It is well suited to locate anomalies despite their depth due to the equalization of the amplitudes of short and long wavelength anomalies (Lahti and Tuomo, 2010). However, the horizontal derivative of the tilt derivative (TDR_THDR), which is invariant of geomagnetic inclination, is capable of providing sharp and well-defined peaks the source edges. Despite being accurate in defining shallow sources it may be ambiguous about deep bodies (Ma and Li, 2012).

### 2.3. Euler deconvolution.

Euler deconvolution is a technique to estimate the depth of the center of the anomalous sources (Thomson, 1982; Reid et al., 1990) based on the x, y and z derivatives of the data and a parameter called structural index (SI) (Cooper, 2008). The SI is related to the geometry of the anomaly sources (Cooper, 2008; Reid et al., 1990). Each Euler solution is calculated based on the data within a window (WS) (Reid et al., 1990) that is recommended to be at least half of the size of the anomaly to compute (FitzGerald et al., 2004). According to the cited literature, SI=0 adequate to define contacts or steps of the causative bodies, whereas SI=0.5 and SI=1 is rather suitable to determine sills and dikes.

### 2.4. Power spectrum.

The spectral analysis of the magnetic data provides information about the depth of the magnetic sources attending to magnetic contrasts (Maus and Dimri, 1995; Peredo et al., 2021). The spectrum pattern is a good approach to estimate the depth of the causative bodies. When plotting the natural logarithm of the spectrum against the radial frequency (or the wavenumber), the different linear segments of the spectrum showed in the graph can be associated with different geological discontinuities. The depth (z) is determined by the slope of each segment (m) according to the formula $z = m/4\pi$ when using power spectrum and lineal wavenumbers (Spector and Grant, 1970; Peredo et al., 2021).

### 3. Geological setting of the study area.

The study area comprises 110184 km2 within coordinates (-2.19º, 43.78º; 3.46º,41,35º) (Fig.1). In addition to the main Pyrenees' zones, some areas at the eastern termination of the mountain range will also be described due to the occurrence of important magnetic anomalies like Olot, La Garrotxa Volcanic field (e.g. Zeyen and Banda, 1988), and the western part of the Gulf of Lion passive margin (e.g. Canva et al., 2020) (see Fig. 1).

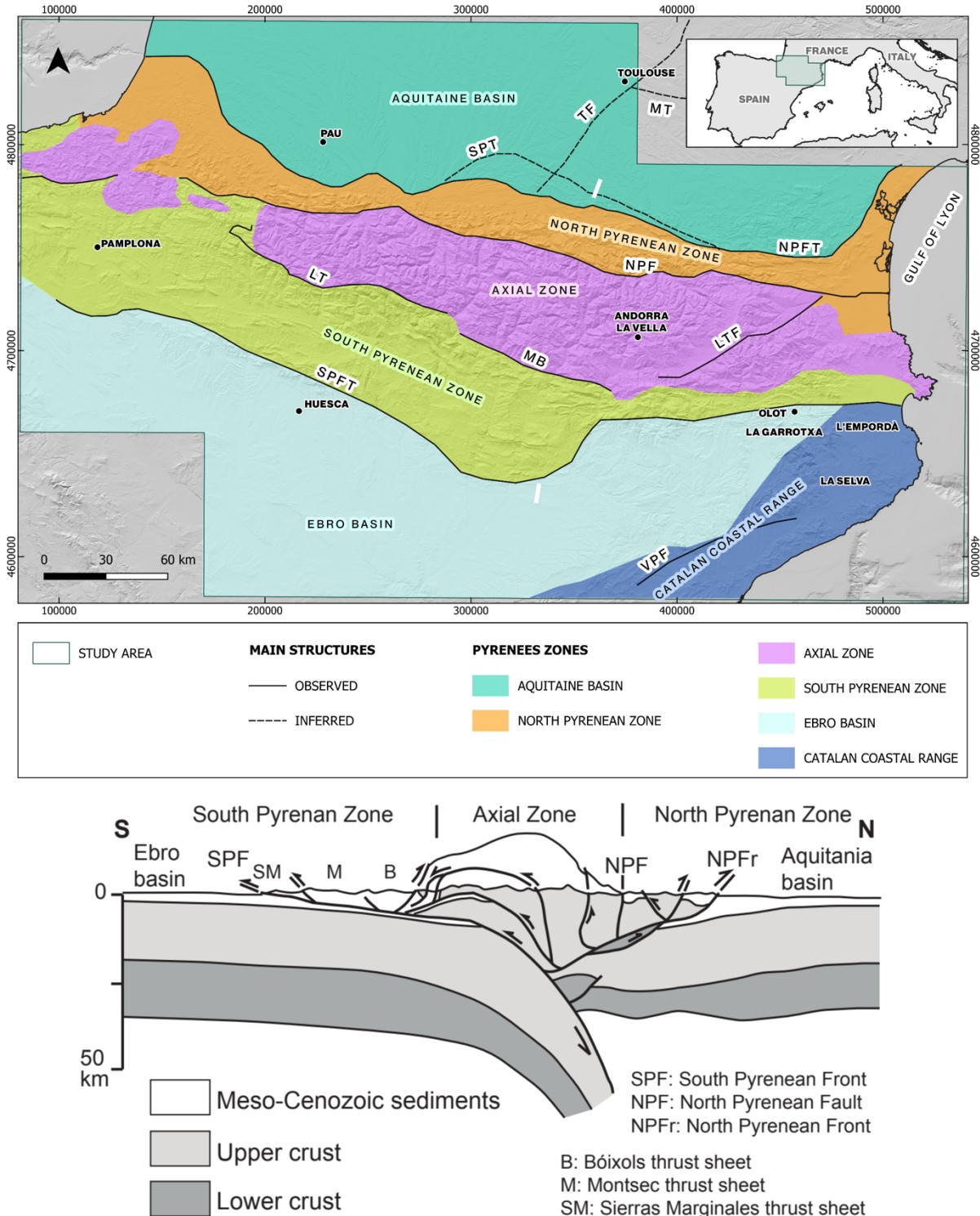

**Fig.1. Geological major units differentiated in the Pyrenees and surrounding areas. Relief map of the study area. Data obtained from the IGN (Instituto Geográfico Nacional, Spain), GEOSERVICE (France) and EMODNET (European Marine Observation and Data Network, Offshore). In the upper right corner, the location of the study area. Below the main map, the ECORS – Pyrenees cross-section. Modified from Muñoz (1992), white lines in the upper map represent the southern and northern approximate boundaries of the profile. NPFT: North Pyrenean Frontal Thrust, NPF: North Pyrenean Fault, LT: Larra Thrust, MB: Morreres Back Thrust, LTF: La Tet Fault, SPT: Sub-Pyrenean Thrust, MT: Mazamet Thrust, TF: Tolouse Fault, SPFT: South Pyrenean Frontal Thrust, VPF: Valles-Penedes Fault. UTM coordinates in m, Zone 31N, ETRS89 datum.**

The Pyrenees represent a WNW-ESE double vergence mountain range with a dominant southward vergence resulted from the convergence between the Iberian and European plates (e.g. Muñoz, 1992). It developed as an

asymmetric chain from Late Cretaceous to the Early Miocene (e.g. Muñoz, 1992) (Fig.2. A). Deformation along the convergent plates boundaries was accommodated by the underthrusting of the Iberian lithosphere underneath Europe and the thrusting and stacking of the upper crust (e.g. Choukroune and ECORS team, 1989; Roest and Srivastava, 1991; Rosenbaum et al., 2002; Wher et al., 2018). The general structure, stratigraphy and geodynamics

of the Pyrenees have been extensively studied during the last five decades (e.g. Séguret, 1972; Garrido-Megías, 1973; Muñoz, 1992; Barnolas et al., 1996; Ford et al., 2022). In parallel, many geophysical studies have been carried out to infer the structure at depth of the Pyrenees, mainly based in seismic, gravimetric and magnetotelluric methods (e.g. ECORS-Pyrenees Team, 1988; Casas et al., 1997; Ledo et al., 2000; Chevrot et al., 2014, 2018; Wang et al., 2016; Campanyà et al., 2011).

The formation of the Pyrenees involved a complex polyphasic evolution, including the reactivation and/or inversion of previous Variscan and Mesozoic structures (e.g., Bond and Mclay, 1995; Poblet, 1991; Muñoz, 1992; Clerc and Lagabrielle, 2012, 2014; Ledo et al., 2000; Wher et al., 2018; Le Maire et al., 2021; Cochelin et al., 2017). The Variscan Orogeny took place during the Carboniferous associated to magmatic activity of different nature (e.g. Zwart, 1986; García-Sansegundo, 1996; García-Sansegundo et al., 2011; Casas et al., 2019) while

different extensional stages occurred from the Permian to Cretaceous, the latter being associated with the opening of the North Atlantic and Bay of Biscay (e.g., Roest and Srivastava, 1991; Ziegler, 1988).

The early stages of Atlantic rifting resulted in the emplacement of subvolcanic tholeiitic dolerites, known as ophites, derived from a subcontinental lithospheric mantle, which are now found as isolated outcrops within the Upper Triassic evaporites in the Pyrenees and other regions of Iberia (e.g., Beziat, 1983; Alibert, 1985). The

Mesozoic extension culminated in the Pyrenees with the formation of a hyperextended margin during the Albian-Cenomanian, characterized by an extreme crustal thinning and subcontinental lithospheric mantle exhumation within the Northern Pyrenees (e.g. Clerc and Lagabrielle, 2014; Jammes et al., 2009; Lagabrielle et al., 2010, Pedrera et al., 2017).

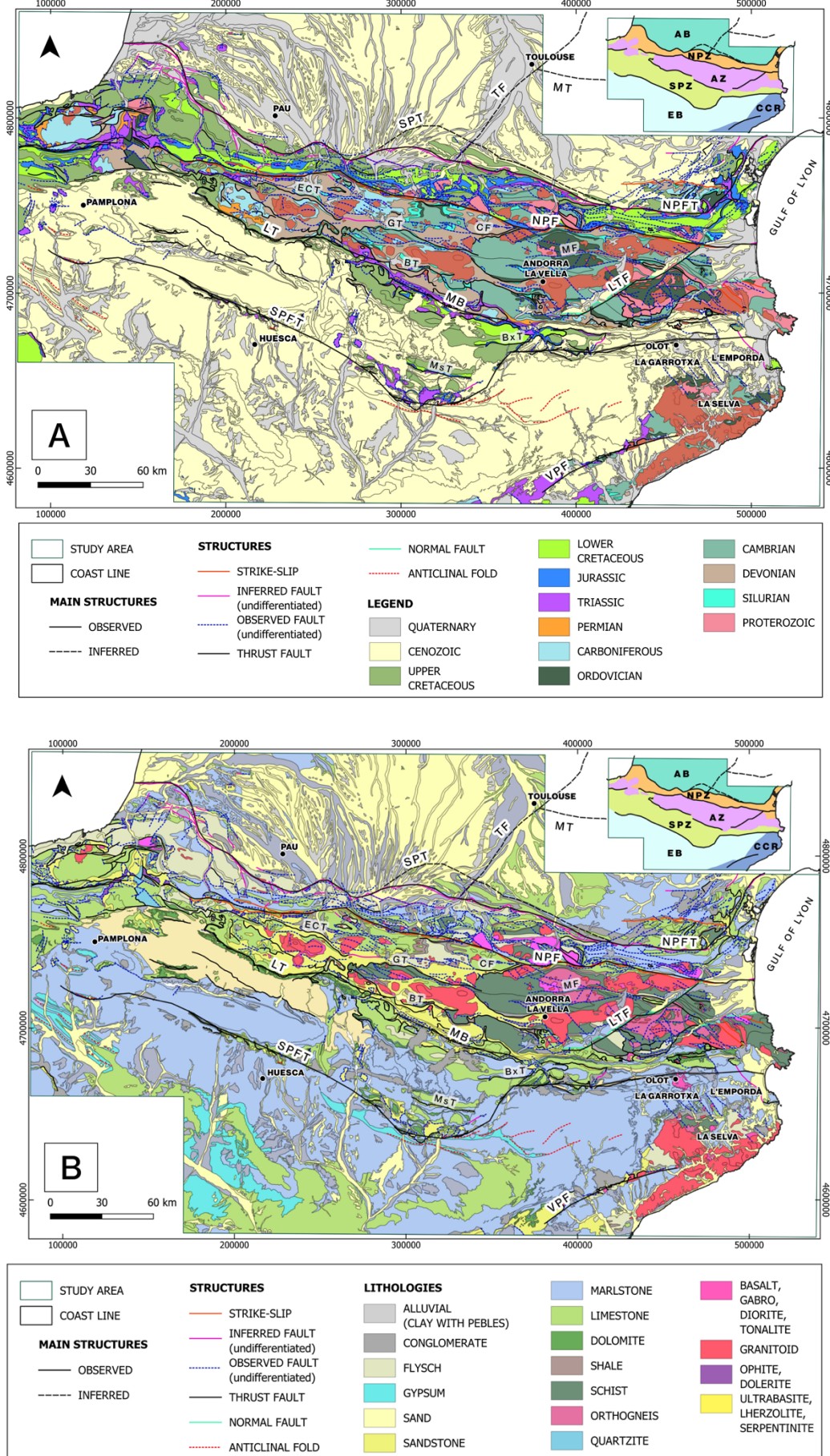


**Fig.2. (A)** Chronological and structural map (Modified from BRGM-IGME, 2009) **(B)** Lithological and structural map (Modified from BRGM-IGME, 2009) of the Pyrenees study area. In the upper right corner, a simplified version of Fig.1. NPFT: North Pyrenean Frontal Thrust, NPF: North Pyrenean Fault, ECT: Eaux-Chaudes Thrust, GT: Gavarnie Thrust, CF: Couflens Fault, MF: Merens Fault, BT: Bono thrust, BxT: Boixols Thrust, MsT: Montsec Thrust, SMT: Sierra Marginales Thrust, LT: Larra Thrust, MB: Morreres Back Thrust, LTF: La Tet Fault, SPT: Sub-Pyrenean Thrust, MT: Mazamet Thrust, TF: Tolouse Fault, SPFT: South Pyrenean Frontal Thrust, VPF: Valles-Penedes Fault. UTM coordinates in m, Zone 31N, ETRS89 datum.

Geological data of the study (age of units, structure and lithology), (Fig.2. A and B) were obtained from the cartography base map of the IGME-BRGM (2009) through the Geological Survey of Spain (IGME-CSIC) website link: http://info.igme.es/cartografiadigital.

### 3.1. Pyrenean main tectonic units.

From North to South, five main tectonic units can be distinguished in the Pyrenees (Mattauer, 1968; Barnolas and Pujalte, 2004 and references therein) (Fig.1); the Aquitaine basin (AB), the North Pyrenean Zone (NPZ), the Axial Zone (AZ), the South Pyrenean Zone (SPZ), and the Ebro basin (EB).

The Aquitaine basin (Fig.2. A and B) represents the northern foreland basin developed on the overriding European plate. It has been extensively explored over the last 60 years (e.g. Biteau et al., 2006; Serrano et al., 2006). Its mid-Eocene to Miocene sedimentary fill can be attributed to a foreland subsidence directly linked to the growth of the Pyrenees (e.g. Biteau et al., 2006; Rougier et al., 2016). Geophysical data, like seismic profiles (eg. Rougier et al., 2016), have revealed the existence of highly deformed Paleozoic basement rocks at depth and Mesozoic sub-basins related to an important rift system highly variable along-strike (e.g. Canérot et al., 2005). The Aquitaine basin is mainly deformed at depth between the North Pyrenean Frontal thrust (NPFT) and the Sub-Pyrenean thrust (SPT).

The North Pyrenean Zone is characterized by inverted thick sequences of syn-rift to postrift Mesozoic rocks (e.g. Souquet et al. 1977; Déramond et al., 1988). This zone represents a relatively narrow fold-and-thrust belt, measuring between 25 to 35 kilometers, which has developed along the southern boundary of the European (upper) plate between the North Pyrenean Fault (NPF) and North Pyrenean Frontal thrust (NPFT) (Fig. 2. A and B). Regional high-temperature low pressure (HT-LP) metamorphism and about 40 outcrops of sub-continental peridotites are hosted in the Northern Pyrenees related to the Early Cretaceous hyper-extension and mantle exhumation (e.g. Jammes et al. 2009, 2010; Clerc et al. 2012, Vauchez et al. 2012).

The Axial Zone (Fig. 2. A and B) is located at the core of the alpine chain and comprises an antiformal stack of south-vergent basement thrust sheets. It mainly consists metasedimentary Paleozoic rocks deformed and metamorphosed previously during the Variscan Orogeny, along with variscan granitoids (e.g. Barnolas et al., 1996; Casas et al. 2019). The Axial Zone elevation varies along its strike and exhibits a direct correlation with the depth of the European basement top (e.g. Soto et al., 2006, Saspiturry et al., 2019) indicating that the control of the deep crustal configuration of the Pyrenees lies on their structural architecture, since the subduction is deeper in the center of the mountain chain and decreases laterally (e.g. Beaumont et al., 2000; Jammes et al., 2009; Saspiturry et al., 2019; Muñoz, 2019).

The South Pyrenean Zone constitutes a south-verging thin-skinned fold-and-thrust belt detached from the basement along the Middle-Upper Triassic evaporites (e.g. Séguret, 1972; Muñoz, 1992). It is characterized by a Mesozoic succession with an along-strike thickness variation and a thick syn-orogenic Cenozoic sequence (e.g. Séguret, 1972; Garrido-Megías, 1973; Cámara and Klimowitz, 1985; Muñoz, 1992; Soto et al., 2002) (Fig.2. A and B).

Finally, the Ebro basin (Fig. 2. A and B) represents the southern foreland basin of the Pyrenees and encompasses a continuous sequence up to 5500 meters of latest Eocene, Oligocene and Miocene continental sediments (e.g. Alonso-Zarza et al., 2002). Its autochthonous sediments are deformed by several folds and thrusts detached along the Eocene evaporites close to the South Pyrenean Frontal thrust.

## 4. Data analysis and results.

### 4.1. Reduction to the pole (RTP).

The reduced to the pole magnetic field was calculated according to the declination and inclination parameters corresponding to the average date of data acquisition 1973/06/15.

The magnetic properties of the different rocks, are crucial for understanding the RTP map. These could include from sedimentary rocks to granulites or lherzolites. Along the study area several magnetic mineral rich, such as magnetite or ilmenite, lithologies outcrop being the most important ones the lherzolites, granulites, basalts and meta-basalts.

This way, as a consequence of the rocks, the resulting map displays a highly defined and located anomalies above the source bodies (Fig.3.A). Magnetic anomalies values range between -50.2 to 83.24 nT, while the mean value is -5.79 nT. It is noticeable that the overlaying of the magnetic anomaly maxima is above the Bouguer anomaly maxima (Pedrera et al., 2017) (Fig.3.B and C). Despite the correlation in some western North Pyrenean Zone anomalies and some gravimetric highs along the Mediterranean coast, the gravimetric map displays a more dulled and homogeneous distribution of anomalies in comparison to the RTP magnetic map.

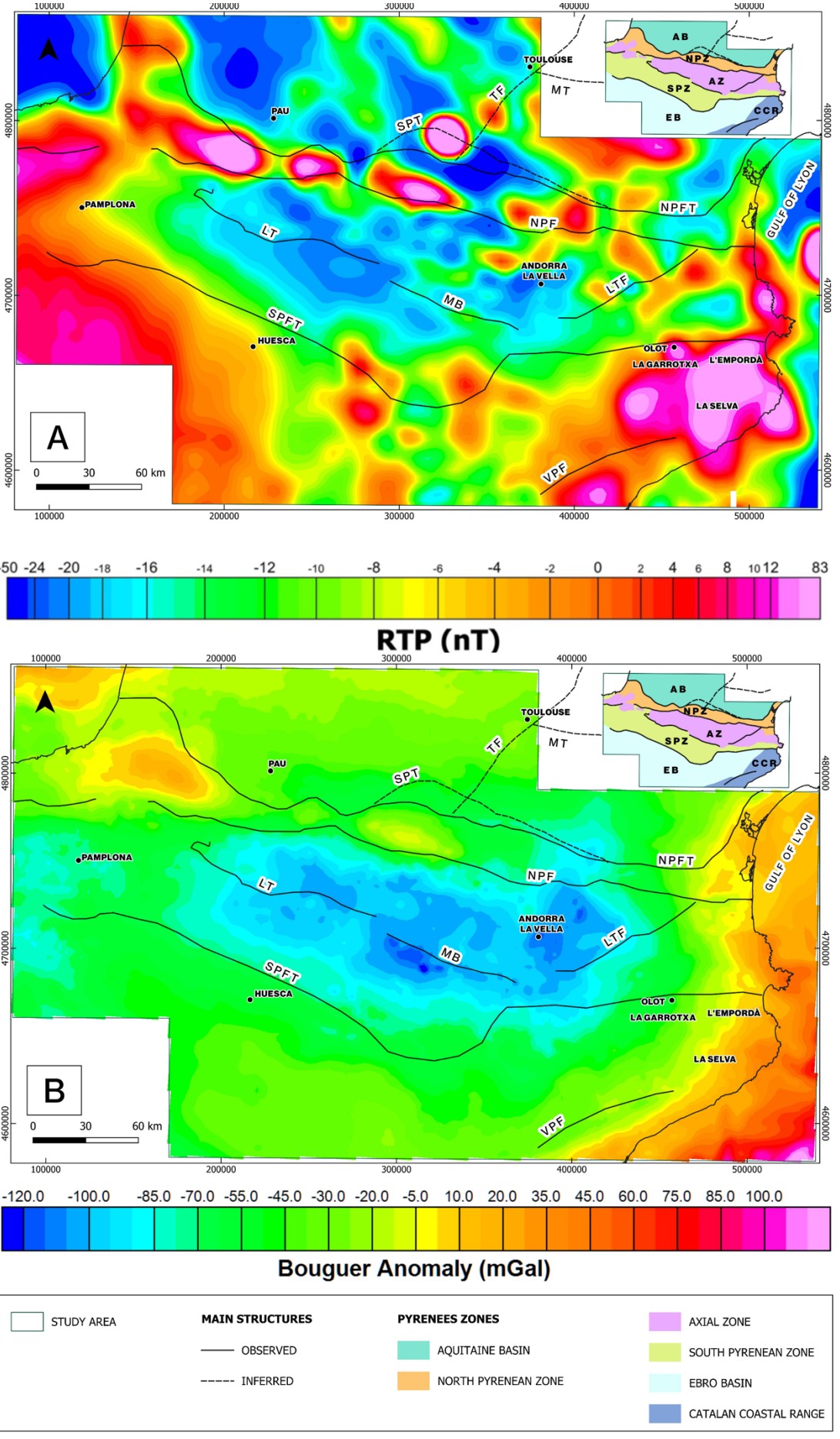

**RTP (nT)**

| -50 | -24 | -20 | -18 | -16 | -14 | -12 | -10 | -8 | -6 | -4 | -2 | 0 | 2 | 4 | 6 | 8 | 10 | 12 | 83 |

**Bouguer Anomaly (mGal)**

| -120.0 | -100.0 | -85.0 | -70.0 | -55.0 | -45.0 | -30.0 | -20.0 | -5.0 | 10.0 | 20.0 | 35.0 | 45.0 | 60.0 | 75.0 | 85.0 | 100.0 |

STUDY AREA

**MAIN STRUCTURES**

—— OBSERVED

- - - INFERRED

**PYRENEES ZONES**

AQUITAINE BASIN

NORTH PYRENEAN ZONE

AXIAL ZONE

SOUTH PYRENEAN ZONE

EBRO BASIN

CATALAN COASTAL RANGE

On the RTP map (Fig.3.A), wavelength values range from 7 to 1000 kilometers. The Pyrenean anomalies can be grouped into magnetic zones that roughly coincide with the main Pyrenean structures.

In the North Pyrenean Zone (Fig. 1, 2. B and 3.A and B), next to the NPFT, the high intensity, short to intermediate wavelength anomalies can be associated with the outcrops of lherzolites, granulites and ophites.

The Axial Zone and South Pyrenean Zone (Fig. 1 and 3. A and B) are characterized by an elongated medium to long wavelength minimum with some short wavelength high amplitude positive and negative anomalies on top. These anomalies could be related to the basement geometry and/or the accumulation of Triassic evaporites. Given that these anomalies have a NNW-SSE direction, they can also be related to Axial Zone structures. In the South Pyrenean Zone is worth noting the so called Lérida anomaly, a medium wavelength high that could be originated by a basement high within the Ebro Basin (Zeyen and Banda, 1988).

Towards the NE, next to the Catalan coast, there is a zone characterized by several maxima that correlates to outcrops of the Olot basaltic volcanic rocks of La Garrotxa and the Volcanic Field of L'Empordà and La Selva. Since these magnetic anomalies are larger than the outcrops of volcanic rocks, this could point to a wider volcanic area than previously thought, with the anomalies also originated subsurface volcanic rocks that could be part of a feeder system. In the Catalan Coastal Range (CCR), the relative maxima are related with outcrops of Paleozoic basement. Towards its northern termination, their anomalies are masked by the magnetic signature of the La Garrotxa Volcanic Field (Olot).

### 4.2. Magnetic derivatives.

Since it focuses on shallower sources, the vertical derivative (VDR_Z) map (Fig.4.A) delimits quite well near surface anomalous bodies. Three different magnetic texture zones are distinguished: a set of NNW-SSE lineaments to the north of the southern border of the Axial Zone; a zone with relatively intermediate anomalies to the Ebro Basin western side; NE-SW lineaments towards the eastern side of the EB, the Catalan Coastal Range and L'Empordà, La Selva and La Garrotxa Volcanic Field.

The horizontal derivative in X (HDR_X) map (Fig.4.B) displays predominantly N-S oriented lineaments along the North Pyrenean Zone, the Axial Zone and the Catalan Coastal Range. These zones also exhibit a strong ESE-WNW component in the horizontal derivative in Y (HDR_Y) map (Fig.4.C), whereas, L'Empordà, La Selva and La Garrotxa Volcanic Field present a more chaotic magnetic pattern.

Horizontal derivatives in N110ºE direction (HDR_110) (Fig.4.D) and N200ºE direction (HDR_200) (Fig.4.E), which are the Pyrenean and its perpendicular main directions, enhance structures in those directions such as La Tet Fault or the Morreres Back Thrust, respectively. HDR_110 seems to be more defined over the Axial Zone and North Pyrenean Zone, and exceptionally eastwards, towards Olot. HDR_200 outlines the southern border of the Axial Zone almost perfectly but has a rather less concordant texture around the other structures.

The analytic signal (AS) (Fig.4.F) shows two different texture zones divided by a NW-SE imaginary line that seems to follow the trace of the Larra and Morreres thrusts, splitting an upper part characterized by rounded shapes above the main anomalies along the North Pyrenean Zone, the Volcanic Field (La Empordà, La Selva, La Garrotxa) and the Gulf of Lyon and a southern part corresponding to the Axial Zone, the South Pyrenean Zone and the Ebro Basin with a diffuse and uneven texture.

Lineaments in the tilt derivative (TDR) map (Fig.4.G) appear to be outlining most of the main anomaly bodies that generate the magnetic anomalies on the RTP map (Fig.3.A). The intricate morphology of the North Pyrenean Zone, the Volcanic Field and the Catalan Coastal Range anomaly sources are very well defined. It is worth noting that from the southern border of the Axial Zone northwards (Axial Zone, North Pyrenean Zone, Aquitaine Basin) it displays a NW-SE texture with a N110ºE approximate direction, while a N-S orientation pattern prevail to the south of that boundary (South Pyrenean Zone, Ebro Basin), being Olot (La Garrotxa) the eastern end of both textures. Its horizontal derivative (TDR_THDR) (Fig.4.H) enhances the edges of the anomaly source bodies, exceptionally along the Central Pyrenees (See section 2.2 Magnetic Derivatives).

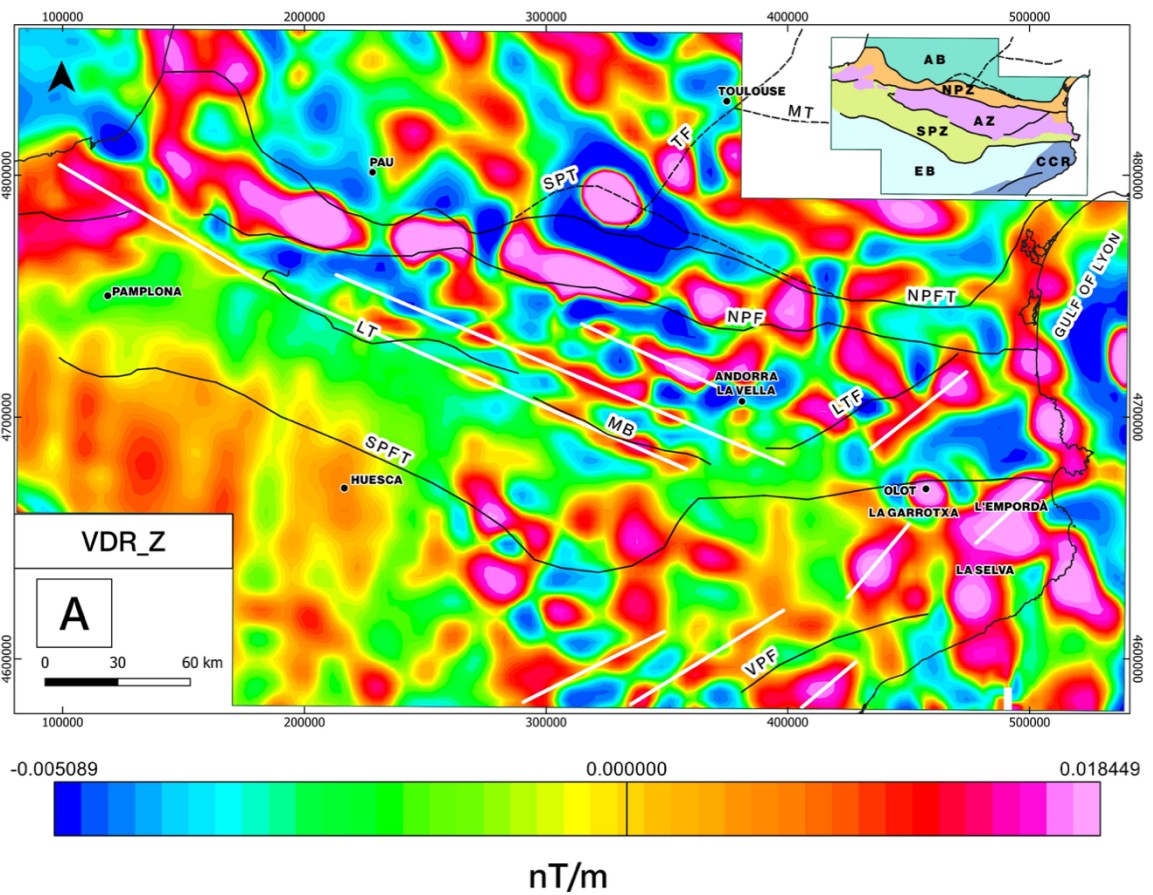

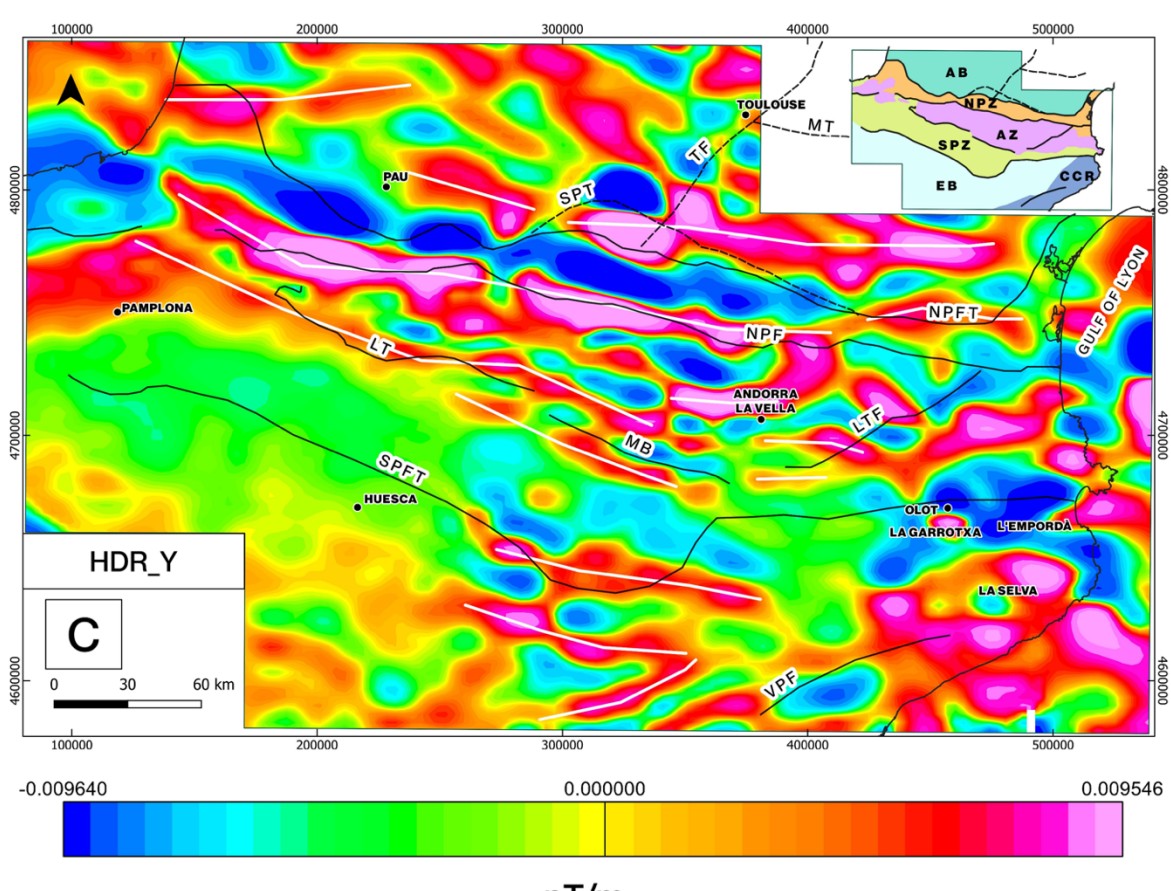

270

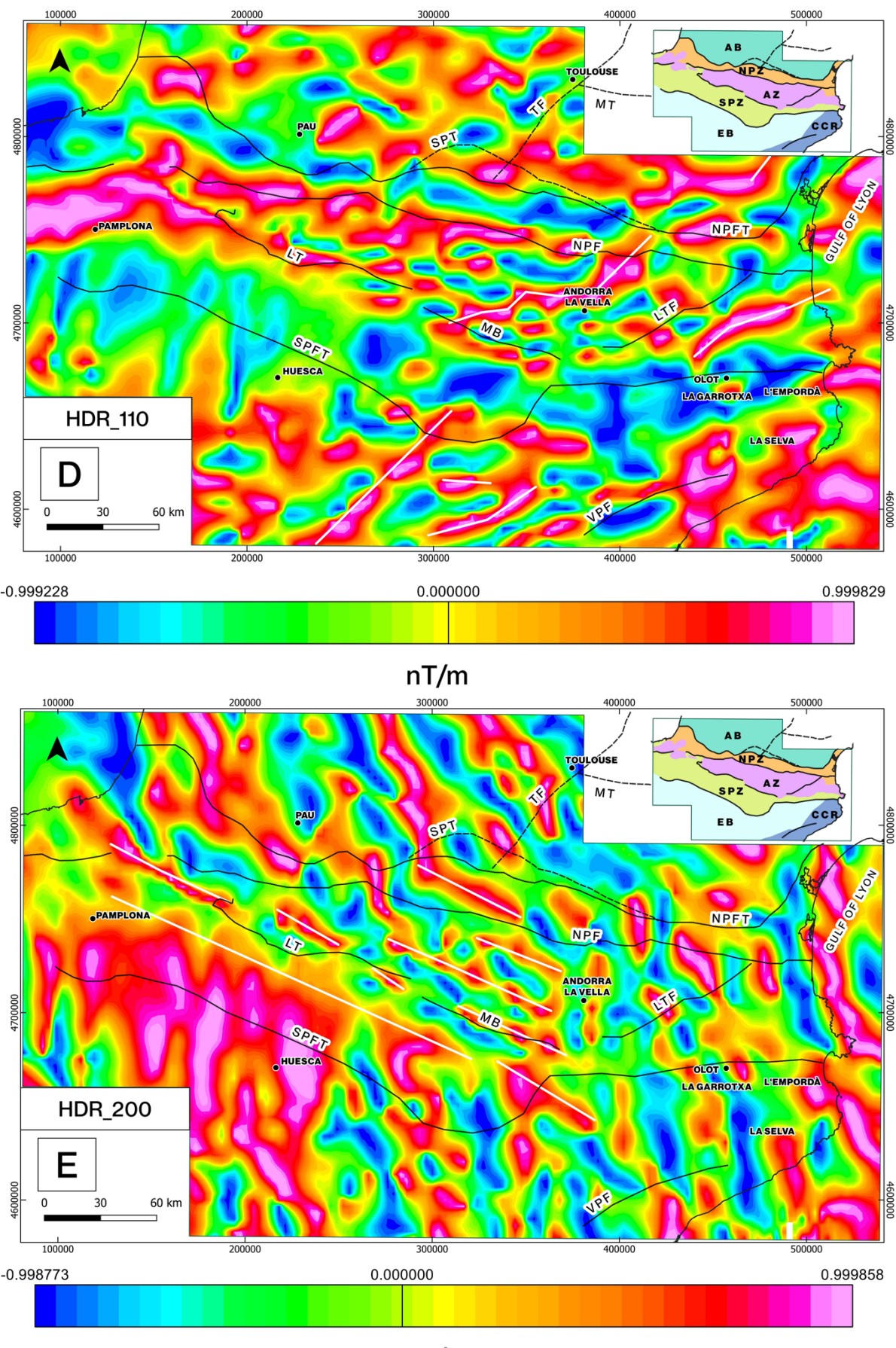

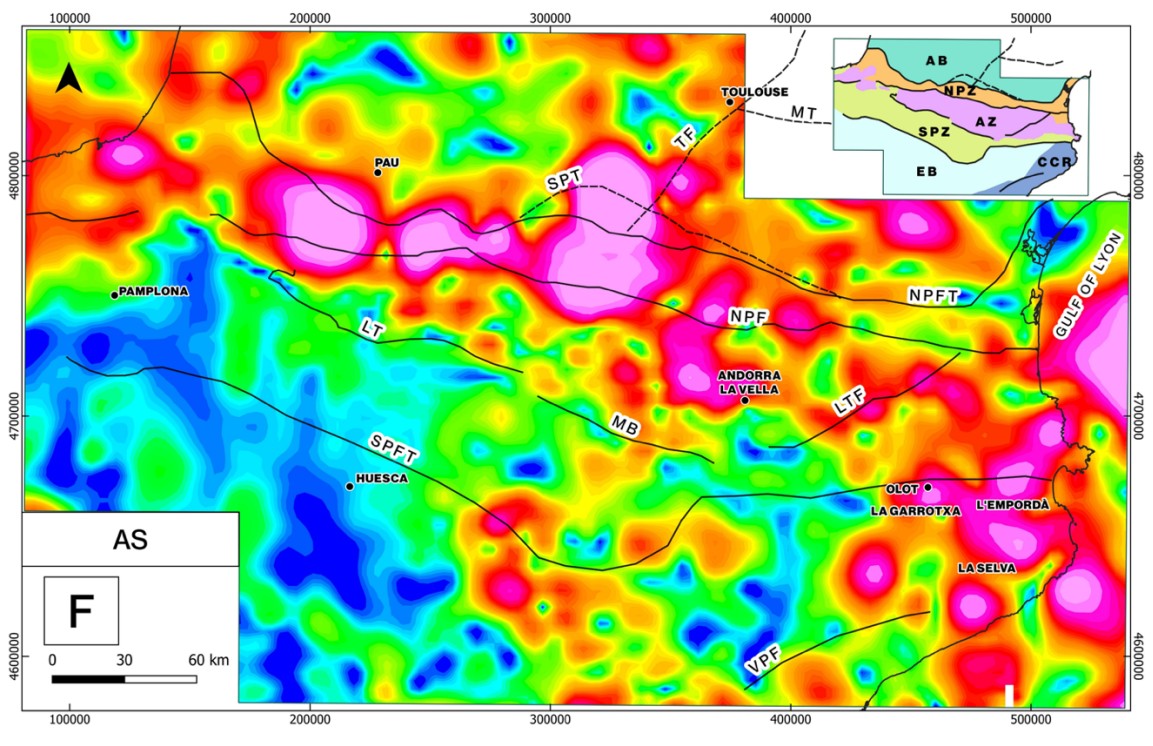

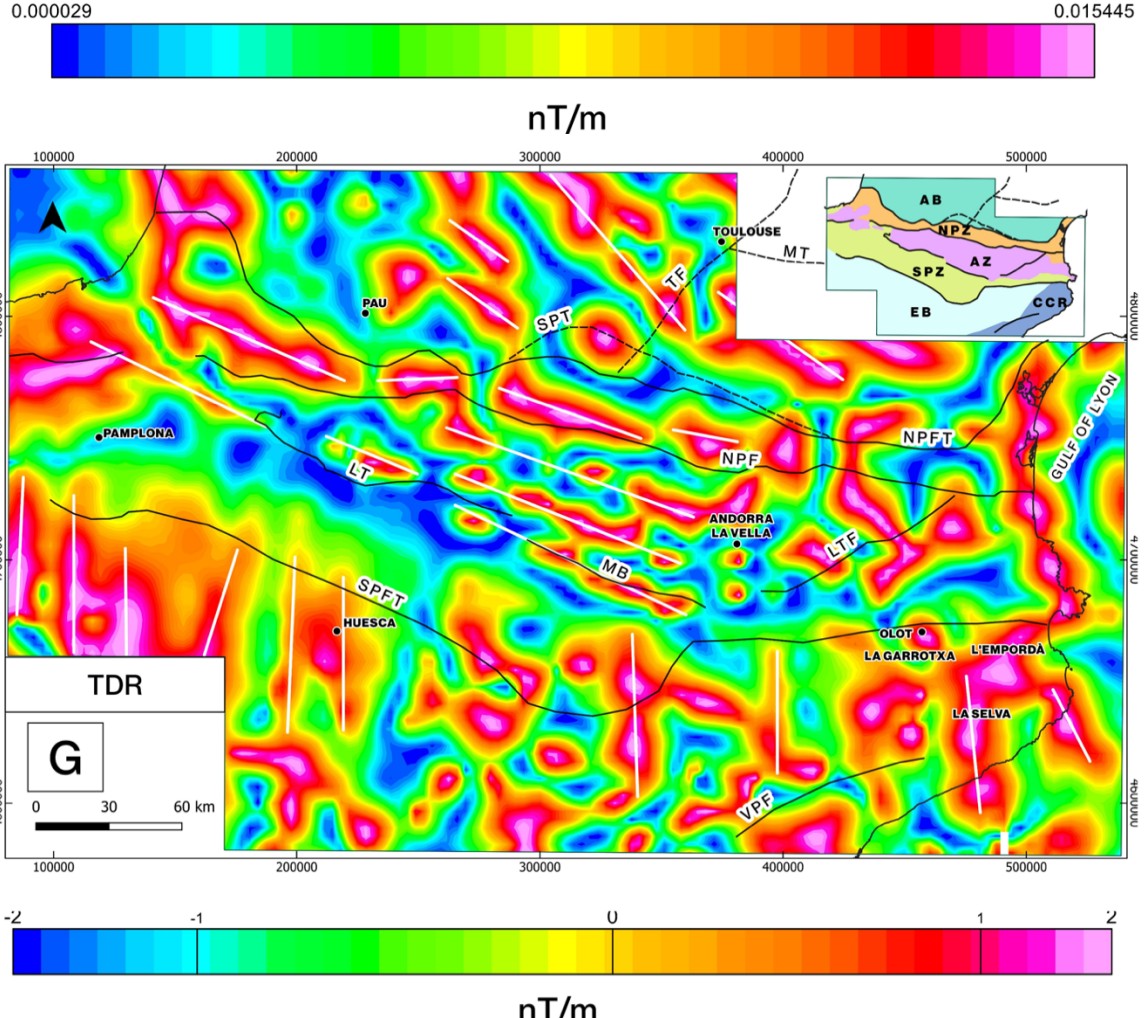

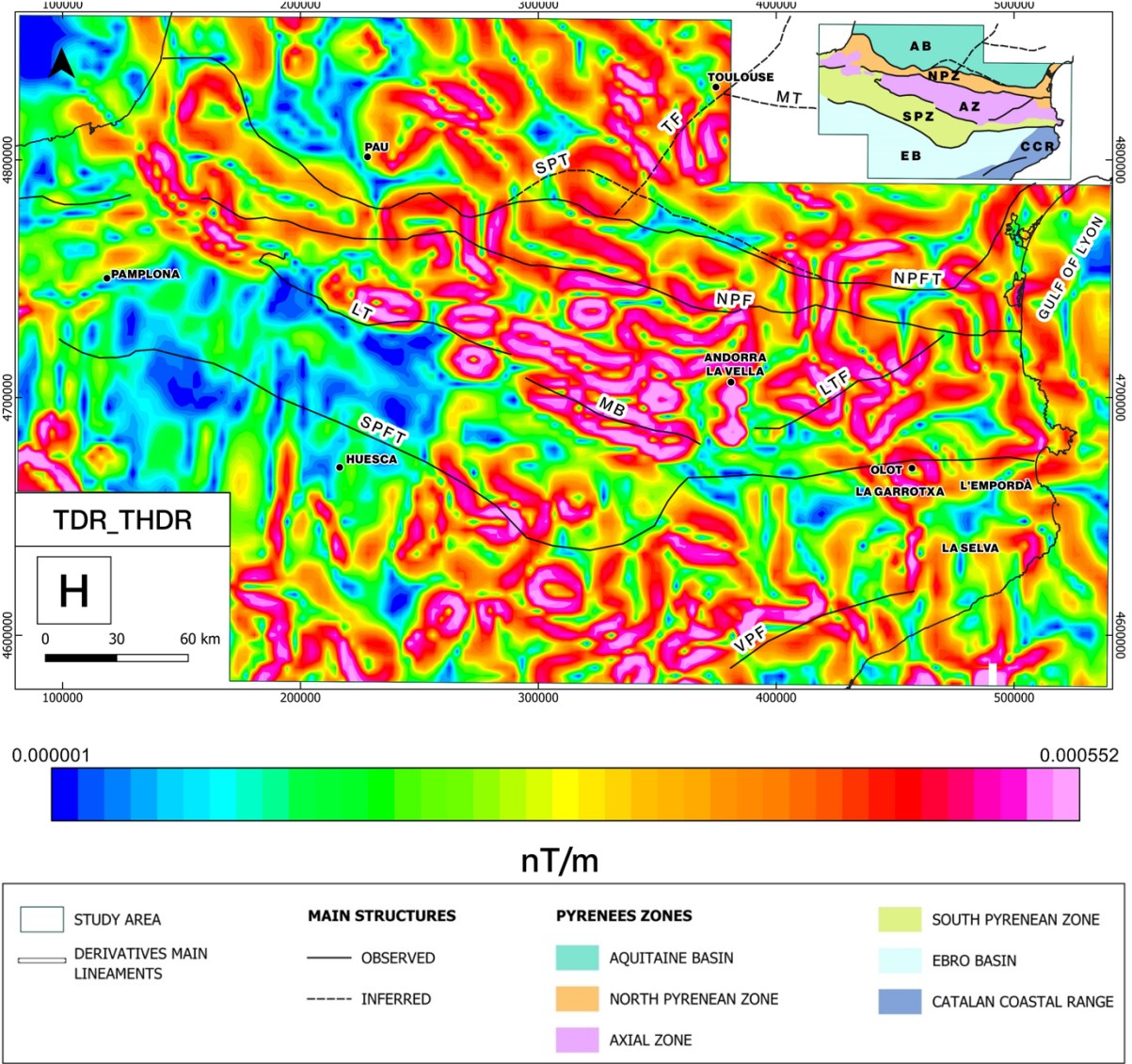

**Fig.4. (A) Vertical derivative map (VDR_Z). (B) Horizontal derivative in X map (HDR_X). (C) Horizontal derivative in Y map (HDR_Y). (D) Horizontal derivative in N110ºE map (HDR_110). (E) Horizontal derivative in N200ºE map (HDR_200). (F) Analytic Signal map (AS). (G) Tilt derivative map (TDR). (H) Horizontal derivative of the tilt derivative map (TDR_THDR). In the upper right corner, a simplified version of Fig.1. In addition, the main lineaments of each derivative and main structures of the study area are also represented. MT: Mazamet Thrust, TF: Tolouse Fault, SPT: Sub-Pyrenean Thrust, NPFT: North Pyrenean Frontal Thrust, NPF: North Pyrenean Fault, LTF: La tet Fault, MB: Morreres Back Thrust, LT: Larra Thrust, SPFT: South Pyrenean Frontal Thrust, VPF: Valles-Penedes Fault. UTM coordinates in m, Zone 31N, ETRS89 datum.**

### 4.3. Euler deconvolution.

In order to constrain the depth estimation of the magnetic source bodies, we have used the Euler deconvolution calculation. As indicated in the methodology, we have chosen the parameters SI=0 (contacts/steps) with a window of 5 km and SI=1 (sills/dykes) with a window of 10 km. Each window size was selected to ensure that the whole extension of the target bodies fit within.

For the contacts/step's solutions (SI=0, WS=5) (Fig.5.A) the depth estimation reaches up to 5 km to the north of the North Pyrenean Fault in different clusters, surrounding some of the RTP high positive anomalies, usually near the outcrops of lherzolites and granulites. The deepest solutions (from 3 km to more than 20 km) are located in

the western sector of the South Pyernean Zone and the Volcanic Field, while shallower solutions (from 0.1 km to 5 km) are found along the Ebro Basin and the Catalan Coastal Range.

For the sills/dyke's solutions (SI=1, WS=10), as the window size increases so does the number of solutions (Fig.5.B). Despite showing a similar distribution as in the previous case, their estimations are deeper than in the previous case. North to the North Pyrenean Fault, the solutions display deeper values to the west than to the east, gathered around the higher value RTP anomalies in the zone. Along the Volcanic Field, deep solutions are found in areas without volcanic outcrops. Depth solutions in the western termination of the Ebro Basin, South Pyrenean Zone, Axial Zone and North Pyrenean Zone (from 10 km to more than 20 km) could be related to a deepening of the top of the basement. Values of 10 km are calculated in the border of the Gulf of Lyon offshore RTP great anomaly. On the other hand, the shallowest estimations (0.1 km or less) are found in the center of the RTP anomaly in the Aquitaine Basin, crossed by the Sub-Pyrenean Thrust (Fig.5.B). However, the outcropping materials do not have a relevant magnetic response in the RTP map (Fig.2.B).

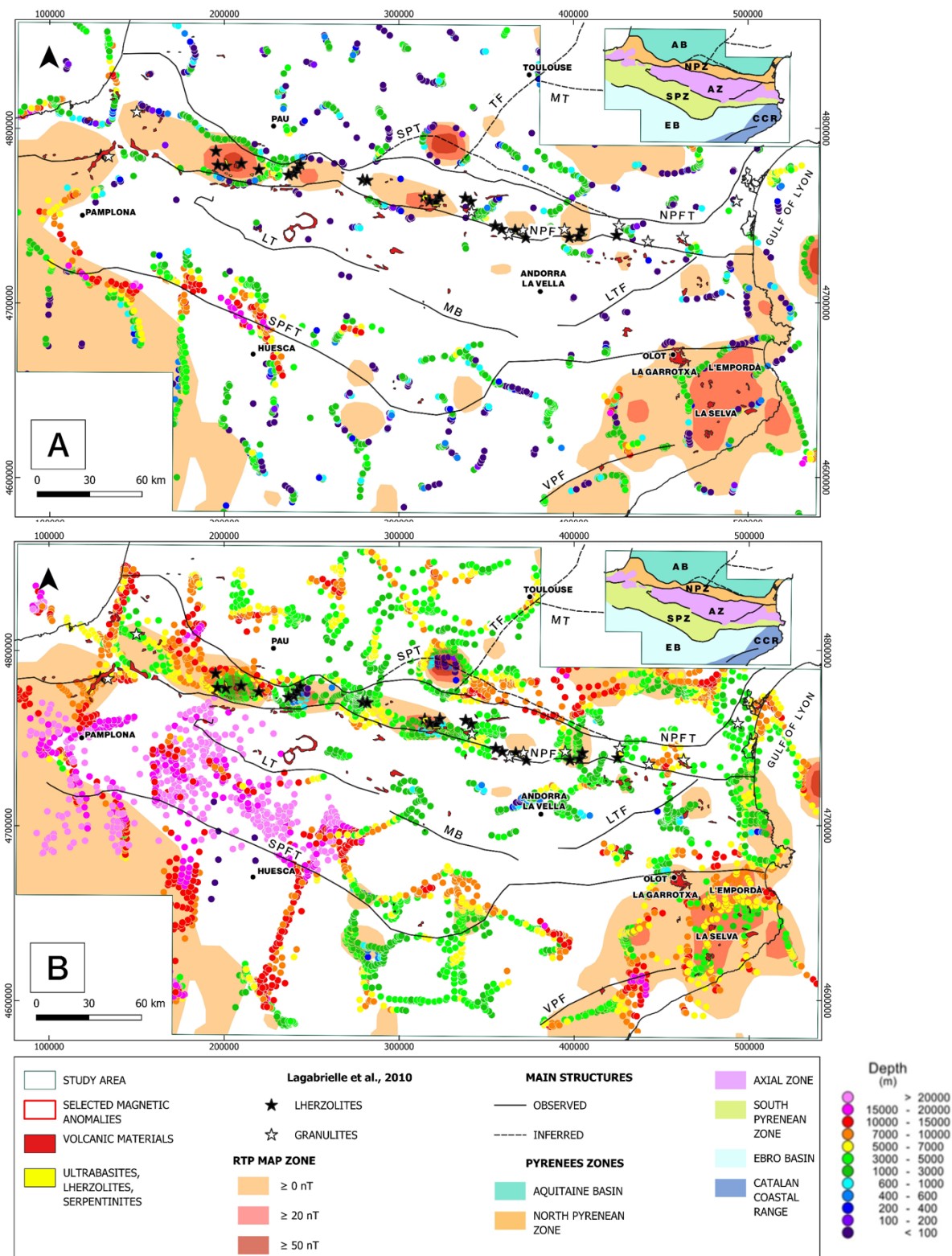

**Fig.5. In order to highlight the most relevant magnetic anomalies, RTP map is only portrayed above 0 nT in addition to (A) Euler solution for SI:0 WS:5; (B) Euler solution for SI:1 WS:10. In the upper right corner, a simplified version of Fig.1. Main structures are also represented. MT: Mazamet Thrust, TF: Tolouse Fault, SPT: Sub-Pyrenean Thrust, NPFT: North Pyrenean Frontal Thrust, NPF: North Pyrenean Fault, LTF: La tet Fault, MB: Morreres Back Thrust, LT: Larra Thrust, SPFT: South Pyrenean Frontal Thrust, VPF: Valles-Penedes Fault. Volcanic materials, lherzolites, Lagabrielle et al., (2009) lherzolite and granulite mapping, Gulf of Lyon, La Selva, L'Empordà, La Garrotxa and Olot are highlighted. UTM coordinates in m, Zone 31N, ETRS89 datum.**

### 4.4. Power spectrum.

The radially averaged spectrum consists mainly in three components (Fig.6) that give the maximum depth of the top of the magnetic basement. A very steep segment (green) that shows a low wavenumber (0.001-0.016 km$^{-1}$). The maximum depth displayed for this segment reaches up to 21-23 km. The next segment (pink) ranges between wavenumbers from 0.016 to 0.084 km$^{-1}$. This segment displays the shallower sources and noise, reaching up to 15 km for the maximum depth. As for the third segment (blue), comprising wavelength values from 0.084 to 0.14

km$^{-1}$ shallower maximum depths are reached despite the peaks associated with noise, around 12 km.

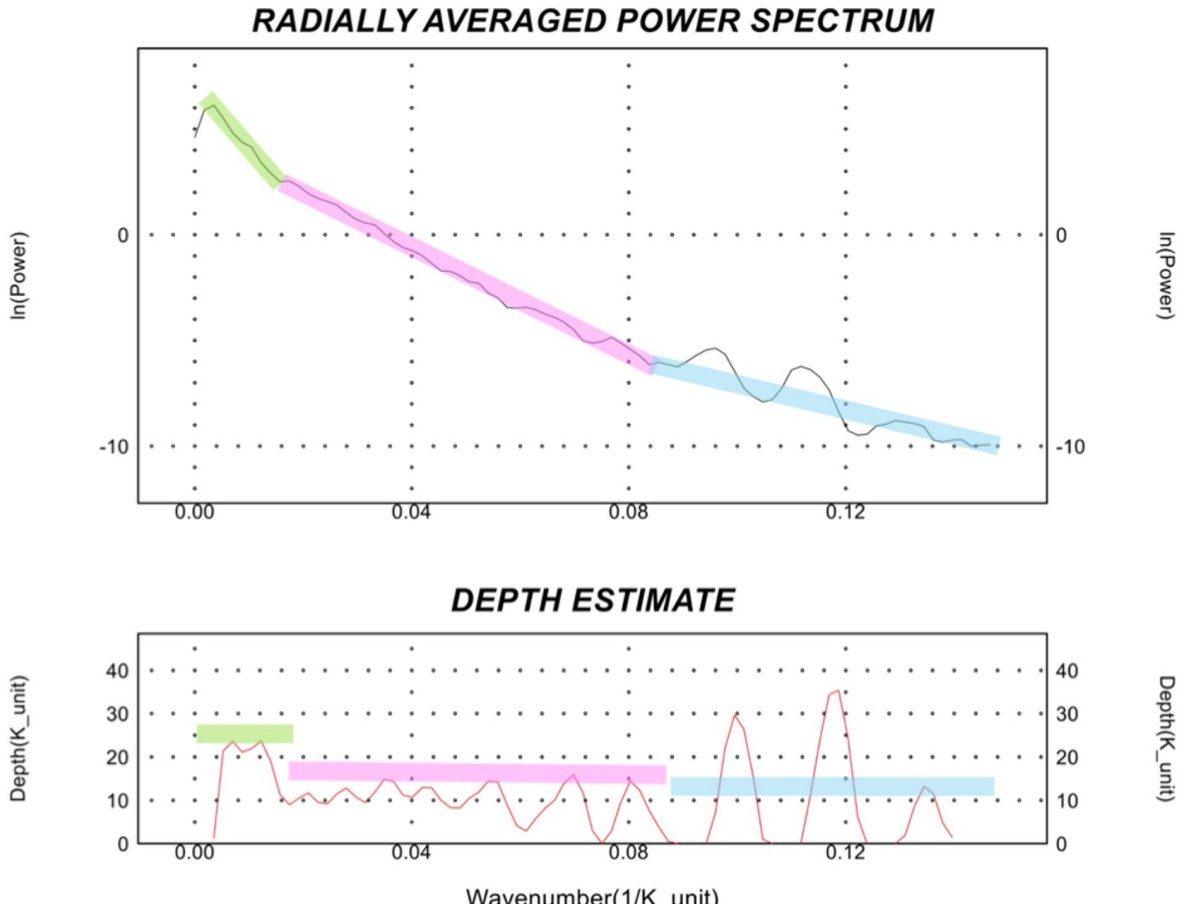

**Fig.6. Radially averaged power spectrum of the RTP magnetic field. The wavelength range spans from 7.14 to 1000 km. The maximum depth values reach 35 km.**

## 5. Structural interpretation of the magnetic data.

### 5.1. Magnetic fabric.

The magnetic texture of the RTP map within the study area is complex and displays a pronounced heterogeneity in terms of source type, distribution and shape (Fig.3.A.B.C). In order to highlight these parameters through the relationship between the magnetic transformations and the geological structures and lithologies, we have built

three merged maps displayed on Fig. 7: (1) A ternary image combining the RTP (Fig.3.A) grid, the Analytic Signal (Fig.4.F) grid and the VDR_Z (Fig.4.A) together with the main lineaments of the HDR_X, HDR_Y, HDR_110 and HDR_200 (Fig.4.B, C, D, E) shown on Fig.7.A since they provide the best approach to the area's structural lineaments; (2) The geological map with the RTP anomalies above 0 nT to enhance the anomalous bodies location and the gravimetric maximums (Fig.7.B); (3) The TDR (Fig.4.G) with the main structures and the

Axial Zone mapping (Fig.7.C).

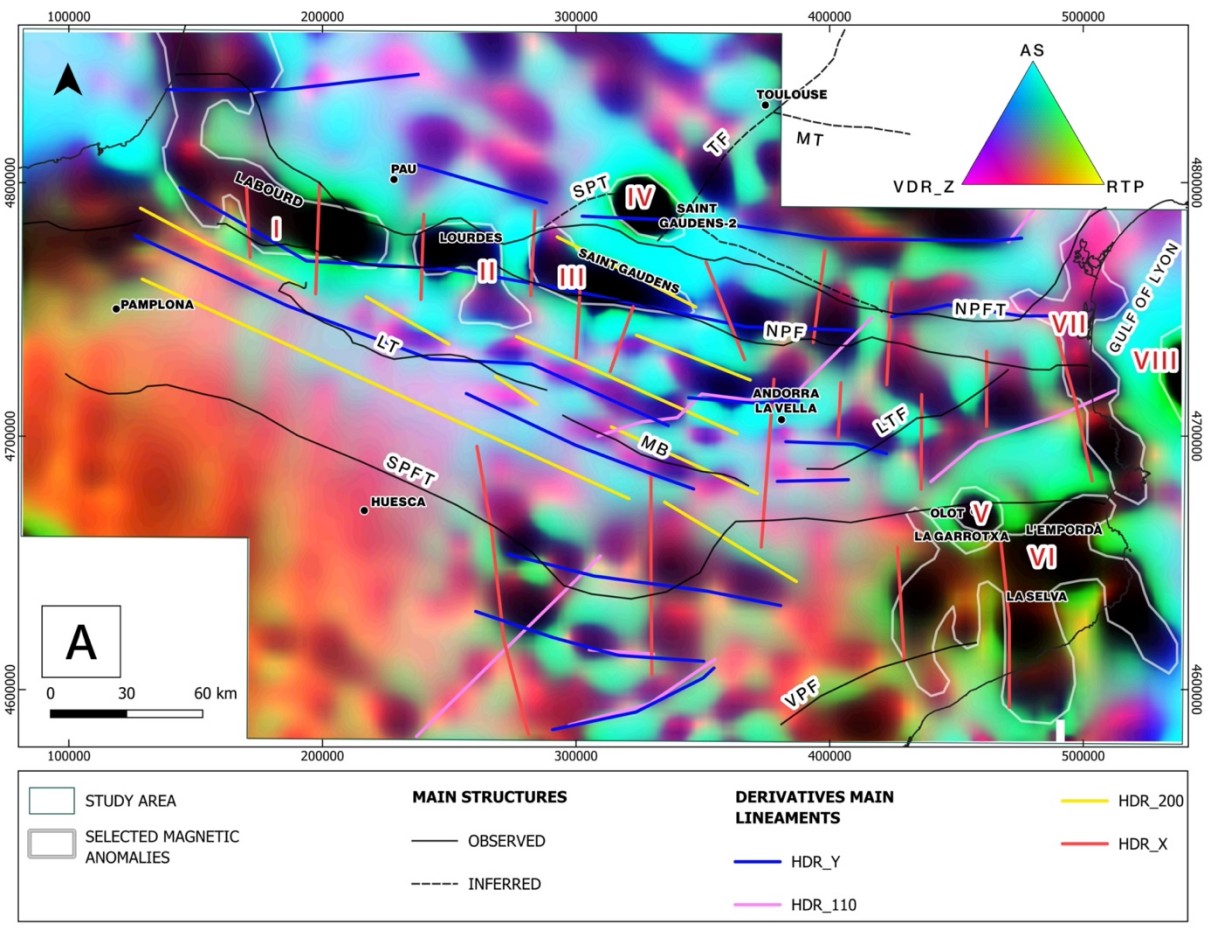

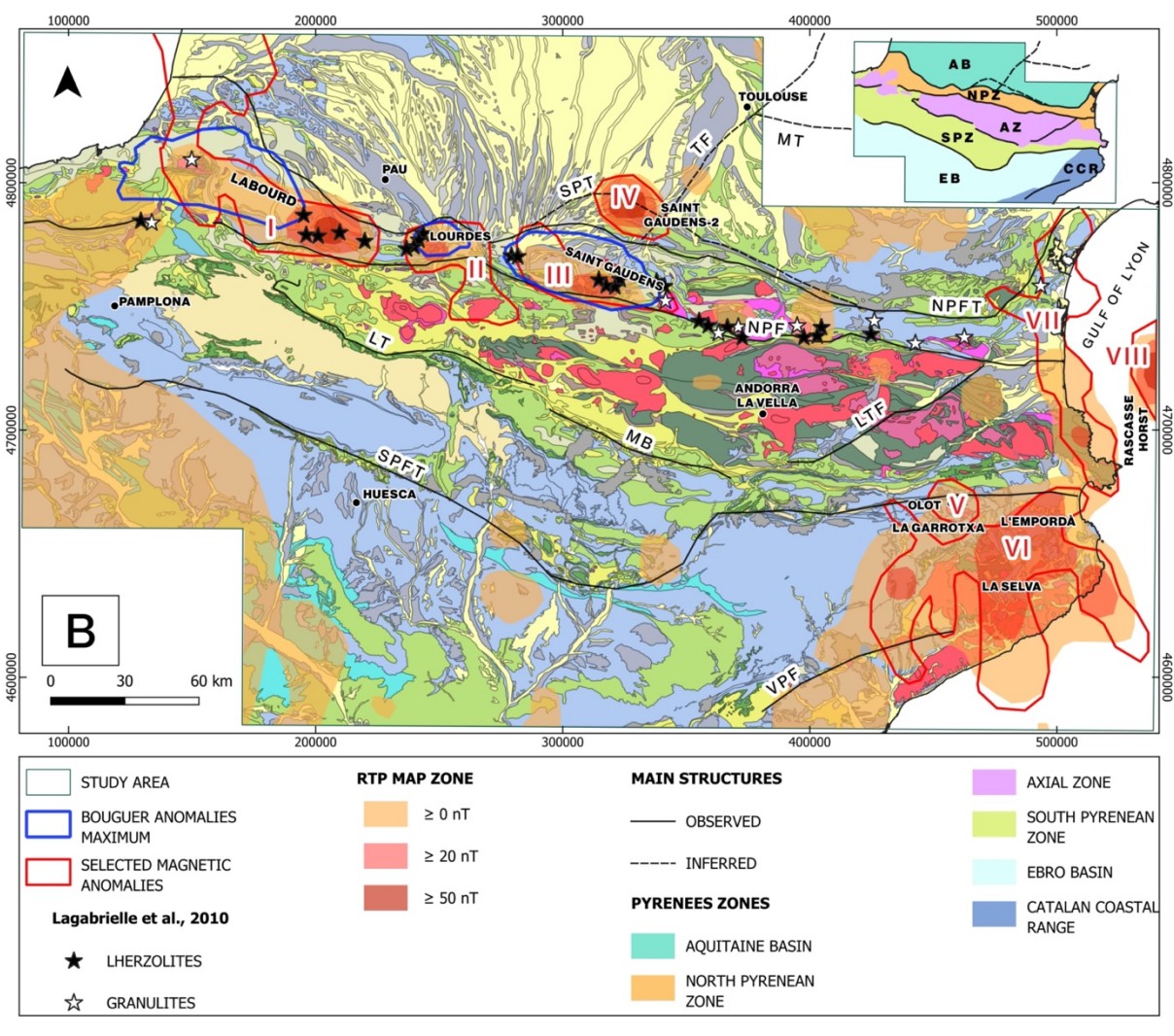

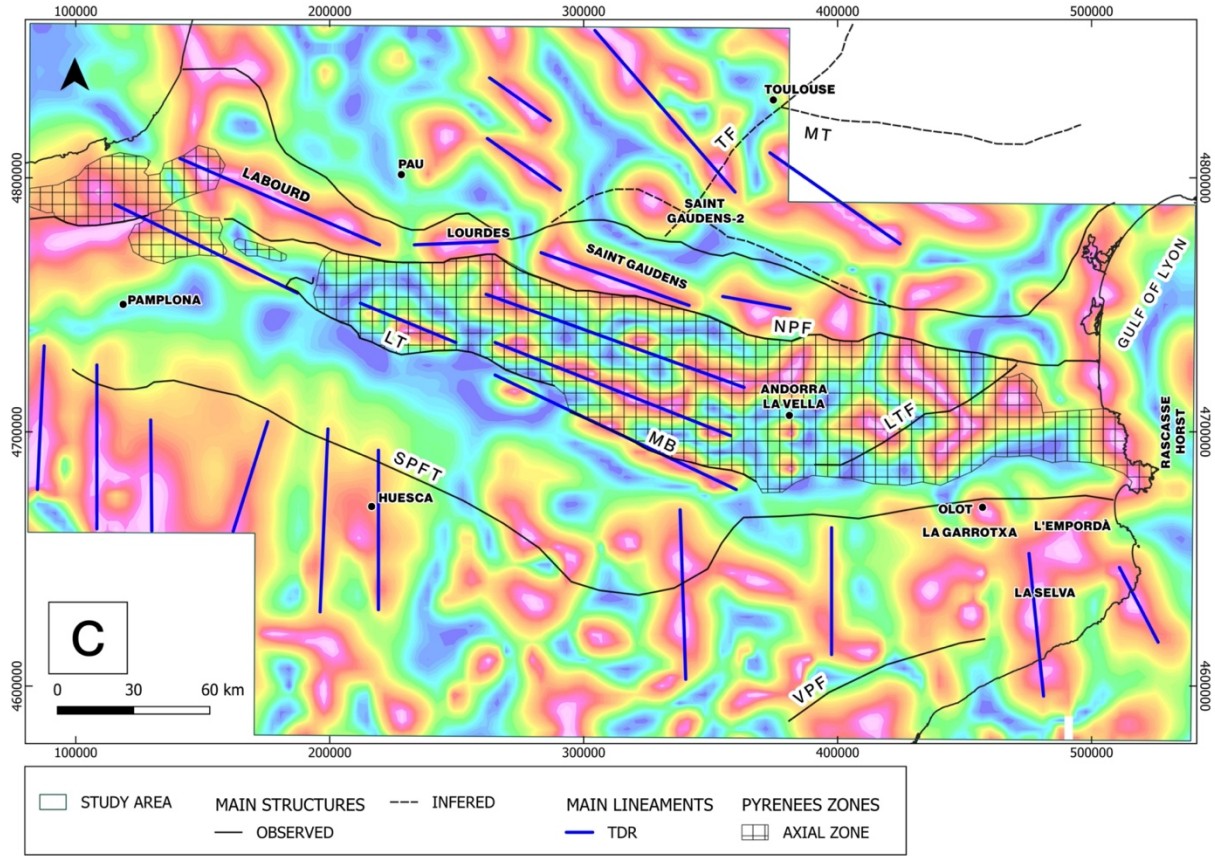

**Fig.7. (A) Ternary image (RTP-AS-VDR_Z) together with the horizontal derivatives lineaments mapping (colour lines) and the selected magnetic anomalies (white lines). In the upper right corner, the triangle represents the color scale. (B) Lithological map of the study area in addition to the reduced to the magnetic pole above 0 nT mapping, gravimetric maximums (blue lines) and selected magnetic anomalies (red lines). In the upper right corner, a simplified version of Fig.1. (C) Comparison between the tilt derivative mapping (blue lines) and the main structures (black lines) with the**
**TDR grid as template and the AZ as a pattern mapping. Main structures represented in black lines for the three maps. In blue: LA: Labourd, L: Lourdes, SG: Saint Gaudens. In red: MT: Mazamet Thrust, TF: Tolouse Fault, SPT: Sub-Pyrenean Thrust, NPFT: North Pyrenean Frontal Thrust, NPF: North Pyrenean Fault, LTF: La tet Fault, MB: Morreres Back Thrust, LT: Larra Thrust, SPFT: South Pyrenean Frontal Thrust, VPF: Valles-Penedes Fault. I: Labourd, II: Lourdes, III: Saint Gaudens, IV: Saint Gaudens-2, V: Olot, VI: L'Emporda-La Selva, VII: East Coast,**
**VIII: Rascasse Horst. UTM coordinates in m, Zone 31N, ETRS89 datum.**

The most outstanding anomaly zones have been selected with the following criteria: Firstly, as shown in Figure 7.A, those areas where high and similar values of RTP, Analytic Signal and VDR_Z coincide, hence displayed in black color, are regarded as most certain evidence of subsurface anomalous bodies. However, their shape is more

accurately constrained by the TDR_THDR peak values (white lines in Fig.7.A), which nearly matches with the shape of the black zones (Fig.7.A). Then, high RTP values (from 20 nT to 50 nT, reaching 70 nT in zone VIII and near 85 nT in zone IV) and the concurrency with Bouguer anomalies (anomalies I, II, III) (Fig.7.B) were also considered for the selection. These anomalies, labelled as I to VIII in Figures 7. A and B, correlate with outcrops of magnetic bodies but also allow delimiting the buried part of these source bodies.

Mapping the main lineations of the horizontal derivatives (X, Y, N200°E and N110°E direction) (Fig.7.A), allows to portrait the magnetic and tectonic fabric. Lineaments with a strong N-S component such as HDR_X and HDR_110 are mostly grouped in the eastern area while those with a predominant E-W component, HDR_Y and HDR_200, are assembled along the Axial Zone and prevail in the western sector. This could imply that the presence of predominantly E-W structures to the west could be linked to the compressive structures related to the

Pyrenees formation (e.g. Muñoz, 1992) while the predominantly N-S structures could be associated to extensive

structures generated as consequence of the opening of the northwestern Mediterranean Sea (e.g. Millard and Mauffret, 2001).

The N-S structures highlighted by the HDR_X lineaments outline the edges of some anomalies that could be related to evaporites to the south-east of the Ebro Basin (Zeyen and Banda, 1989) while the eastern most ones

next to the Catalan Coastal Range seem to be affected by the volcanic field of L'Empordà, La Selva and La Garroxa. This N-S lineation may be related to the volcanic materials disposition in depth since the most notorious anomalies linked to the volcanic materials appear to the east of the lineation while the western side appears to be less magnetic (Fig.7.A and B), separating La Garrtotxa to the west and L'Empordà and La Selva to the east. This layout may be related to a fault system setting that places shallow bodies to the east of the lineaments, whereas

western magnetic source bodies may be located in structurally deeper positions. Additionally, another N-S lineation seems to outline the northwestern edge of the Catalan Coast anomaly and could be linked to the intrusion of the magnetic source materials (anomaly VII). Regarding to the N110ºE direction lineations (Fig.4.D and 7.A), the southern-central ones appear to constrain the evaporite bodies alongside the HDR_X and HDR_Y lineaments. At both sides of La Tet Fault (LTF), lineations with N110ºE direction are mapped (Fig.7.A), which we interpret

that may imply the presence of similar structures along the eastern end of the Axial Zone.

On the other hand, the lineations with an E-W component (HDR_Y and HDR_200) constrain the configuration of the Axial Zone in its western and central parts, where the thickness of the Pyrenean crustal prism is greatest (e.g. Soto et al., 2006; Pedrera et al., 2017). HDR_Y lineations mimic the pathway of the North Pyrenean Fault to the north, as well as the Larra Thrust and the Morreres Back Thrust to the south. When combined with the

HDR_200 lineaments, they outline the anomaly of the southern border of the Axial Zone as well as the morphology of the Saint Gaudens anomaly (III). To the south, they define a parallel texture which may be related to the presence of evaporites (e. g. Santolaria et al., 2020).

Lastly, the tilt derivative peak mapping exposes the pathway of the structures such the North Pyrenean Frontal Thrust, the Morreres Back Thrust and the Larra Thrust, which delimit the Axial Zone (Fig.7.C). In addition to

describing these main structural features, this method portrays structural fabric inside the Axial Zone. A N200ºE lineament domain is displayed westward that turns into a N110ºE eastwards, adjacent to the La Tet Fault (Fig.7.A and B).

### 5.2. Geological interpretation of the positive magnetic anomalies.

The correlation between the magnetic response of the most remarkable magnetic anomalies (Fig.8.A and B) and

their geological context (Fig.2.A) can be interpreted as follows.

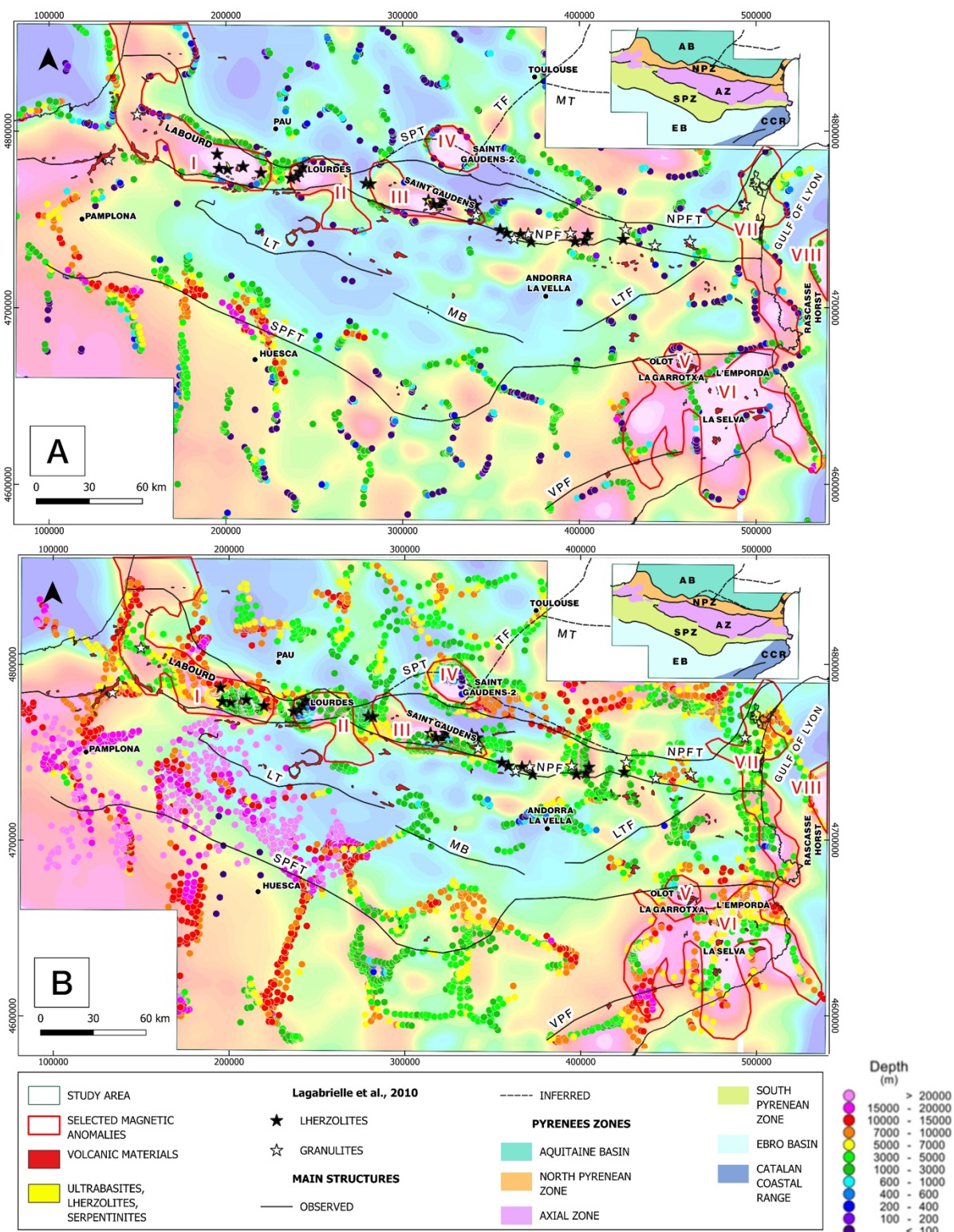

**Fig.8. Reduced to the pole magnetic map of the study area with the (A) Euler solution for SI:0 WS:5; (B) Euler solution for SI:1 WS:10 on top. In the upper right corner, a simplified version of Fig.1. Main structures are also represented. MT: Mazamet Thrust, TF: Tolouse Fault, SPT: Sub-Pyrenean Thrust, NPFT: North Pyrenean Frontal Thrust, NPF: North Pyrenean Fault, LTF: La tet Fault, MB: Morreres Back Thrust, LT: Larra Thrust, SPFT: South Pyrenean Frontal Thrust, VPF: Valles-Penedes Fault. Volcanic materials, lherzolites, Lagabrielle et al., (2009) lherzolite and granulite mapping are displyed. I: Labourd, II: Lourdes, III: Saint Gaudens, IV: Saint Gaudens-2, V: Olot, VI: L'Emporda-La Selva, VII: East Coast, VIII: Rascasse Horst.**



### 5.2.1. Aquitaine Basin.

Located in the southern border of the Aquitaine Basin, the anomaly IV has previously been interpreted by Le Maire et al. (2021) as linked to a slice of sub-continental mantle. This magnetic anomaly does not coincide with lherzolite or granulite outcrops (Fig.2.B and 7.B) and it is associated with a mild gravimetric response (Fig.3.B and C), it may correspond to a portion of serpentinized mantle. Nonetheless, this magnetic anomaly coincides with the inferred trace of the Sub-Pyrenean Thrust (SPT) (Angrand et al., 2018).

In the RTP map, Anomaly IV displays a magnetic response up to + 83 nT, the highest in the study area. According to Euler solutions, it exhibits the shallowest depths for SI=0 (0.1 to3 kilometers) suggesting the presence of a body at those depths (Fig.5.A and 8.A). SI=1 shows a well-shaped circular geometry with average depth up to 7 km (Fig.5.B and 8.B. According to subsurface data, Rougier et al. (2016) already interpreted, coinciding with the trace of the Sub-Pyrenean Thrust (SPT), the presence of Upper Albian-Cenomanian volcanic rocks at depth, which may have used previous normal faults as a conduit. Therefore, we interpret magnetic anomaly IV as related to a highly magnetic and low-medium density vertical volcanic dyke-like body as the source.

### 5.2.2. North Pyrenean Zone.

The North Pyrenean Zone is characterized by the presence of magnetic anomalies I, II and III. Their trace match exceptionally well with the outline of the gravimetric anomalies (Fig.7.B), also located between the NPFT and the NPF. The shape of these anomalies displayed in the RTP map is a WNW-ESE oriented elongated shape (Fig.3.A), however the ternary image (Fig.7.A) suggest a more uneven shape that could reveal the source extension at greater depths.

Regarding the Euler solutions, these anomalies are not well defined for SI=0. For SI=1, the deep solutions are found approximately at the edges of the anomalies (up to 15 km) commonly between 1 and 5 km and reaching more than 20 km for the northwestern areas of anomaly I (Fig.8.A and 8.B). Whereas, for SI=1 the shallow depths are registered at the center of the clusters (up to 1 km) (Fig.5.B and 8.B).

Labourd anomaly (I) appears to be continued towards the NW, probably deepening as depicted in the Euler Solutions (SI=0 and SI=1) (Fig.8.A and B). Lourdes anomaly (II) seems to be connected to a southern small WNW-ESE oriented body that appears to be at an intermediate depth (1 to 3 km) from the Euler solutions (Fig.8.A and B). Saint Gaudens anomaly (III) has an elongated shape that, in the Euler Solutions, is delineated by a string of solutions with depth ranging between 0.6 and 7 km.

From a geological point of view, the North Pyrenean Zone is characterized by the presence of several outcrops of lherzolites, granulites and serpentinites (see the mapping compilation of Lagabrielle et al., 2010) that have been already linked to magnetic anomalies I, II and III (e.g. Casas et al., 1997; Le Maire et al., 2021), as their outcrops cover the extension of these anomalies (Fig.7.B). These rocks consist of basic and ultrabasic mantle and lower crustal rocks emplaced at upper crustal levels related to an extreme crustal thinning and subcontinental lithospheric mantle exhumation occurred within the Northern Pyrenees during the Albian-Cenomanian (e.g. Clerc and Lagabrielle, 2014; Jammes et al., 2009; Lagabrielle et al., 2010).

### 5.2.3. Axial Zone.

As shown in Fig.1.A and 3.A, the Axial Zone does not exhibit any remarkable anomalies along the Pyrenees Mountain Range (anomalies range between −20 to 0 nT). The absence of highly magnetic bodies derived from the mantle is consistent with the structural architecture of the Axial Zone, formed by a thick alpine stack of basement thrust sheets. Therefore, in contrast to the North Pyrenean Zone, Axial Zone magnetic mild response

may be on account of an increase of the upper crustal thickness and the fact that the granitoids and other intrusive rocks have very low magnetic susceptibility, most of them falling on the paramagnetic domain (Pueyo et al., 2022; Porquet et al., 2017). Some of the WNW-ESE magnetic lineaments present at the western side of this zone could be related to the Variscan deformed and elongated metamorphic domes (Cochelin et al., 2017).

Nevertheless, the anomaly VII, located at the eastern coastline, present values of at least + 20 nT. Its trace follows northward from the main RTP anomaly, as portrayed in the ternary image (Fig.7.B), and represents the transition zone between the Axial Zone, the Mediterranean coast and the westernmost Gulf of Lyon. The outcrops of Cambrian meta-basaltic rocks (Debon et al., 1996; Padel et al., 2018), as well as Pleistocene volcanic rocks buried which show small outcrops in this area may be the anomaly source (Carreras & Druguet, 2014, Carreras et al., 1988).

Zone VII is not well defined in the Euler Solutions. The values of the solutions (Fig.8.A and B) are shallower for SI=0 (up to 5-7 km) and deeper for SI=1 (up to 7-10 km), yet in both cases the source of the anomalies deepen towards the north.

It is significant to highlight the existence of low intensity anomalies that have the same orientation as the LTF to its west (Fig.3.A) as well as the northern area of anomaly V, which reaches depths of 3-5 km, probably indicating the presence of a bigger spread of volcanic materials beneath the surface (Fig.7.B).

**5.2.4. South Pyrenean Zone.**

The RTP map (Fig.1.A and 3.A) reveals a rather low magnetic response for the SPZ. Its south-central area displays values up to 6 nT (Fig.7.B). this area, small and dull responsive magnetic anomalies have been linked to the presence of underlying Triassic ophites by Zeyen and Banda (1989), but we do not have enough data to confirm this hypothesis. The Euler solutions of SI=0 mainly follow the SPFT with depths that can reach up to 2 km westwards. By contrast, SI=1 describes two distinct areas, a deeper area (7 to 20 km) in the western half and a shallower area (0.2 to 10 km) to the east (Fig.8.A and B). Magnetic anomaly sources within the area located at depths below 2-6 km (SI=1) might be linked to sources located in the footwall of the South Pyrenean Thrust according with the geometry interpreted of Pyrenean crustal cross sections (see Teixell et al., 2018)

**5.2.5. Catalan Coastal Range.**

Characterized by high positive anomalies (20-50 nT) in the RTP map (Fig.1.A and 3.A), the CCR structure is also highlighted by the derivatives (e.g. Fig.4.G) and the ternary image (Fig.7.A). These are linked to the anomalies V and VI on Fig.7.A and B.

Anomaly V, located around the town of Olot correspond to the outcrop of basalts and basanites of the La Garrotxa volcanic unit (Zeyen et al., 1991). This Neogene and Quaternary volcanic materials are genetically correlated to those of anomaly VI. The latter compromises the L'Empordà and La Selva volcanic units which are mainly Neogene (Dèzes et al, 2004, Araña et al. 1983, Zeyen et al., 1991). These volcanic terrains that extend offshore involve few volcanic outcrops, however, there is a great amount of granitoids and intermediate to ultramafic plutonic rocks outcropping (Enrique, 1990). We interpret the high values of RTP anomaly (up to 50 nT) as related to the underlaying basalts and outcrops of ultramafic rocks.

For the Euler solutions, anomaly V seems to display shallow solutions towards the edges for SI = 0 ($<0.1 - 0.2$ km), reaching 1-5 km for SI=1 (Fig.8.A and B). Anomaly VI values go around 1-5 km for the SI=0 (Fig.8.A) around the edges and from 5-10 km for the buried volcanic materials for SI=1 (Fig.8. B), reaching up to 20 km in the westernmost areas.

### 5.2.6. Ebro Basin.

The RTP map displays anomaly values above 0 nT with a mainly N-S orientation for most of the area, mostly on its western and eastern edges, exceptionally in the center, extending from those previously described to the south of the South Pyrenean Zone. The central anomalies located in the South Pyrenean Zone seem to extend southwards, showing the lowest magnetic anomaly values across the area. However, westward within the area, a high anomaly value (+10 nT) appears gradually portraying the northern edge of the Iberian Chain (Fig.1.A and 3.A) that could be related to a shallower top of the basement.

### 5.2.7. Gulf of Lyon.

Anomaly VII represents the abrupt eastern end of the Pyrenees Mountain Range as the magnetic fabric changes drastically in the North-Western Mediterranean Sea (Canva, A. et al., 2020). In addition to the contribution of the Cambrian metabasalts (Debon, et al., 1996; Padel et al., 2018), this anomaly is suggested to be related to the Oligocene-Miocene rifting that opened the Western Mediterranean Sea with a progressive thinning of the crustal thickness from the Pyrenees to the Mediterranean Sea (Mauffret et al., 2001) that also generated the Pleistocene Volcanic rocks. The presence of the Catalan Transfer Zone as well as other NW-SE structural lineations, crucial during the opening Gulf of Lyon, (Canva, A. et al., 2020) may be genetically linked to the NW-SE magnetic anomalies, such as VIII.

This anomaly comprises the Rascasse Horst, part of a NE-SW horst-graben display of thinned continental crust (Gorini, 1993; Séranne et al., 1995; Gailler et al., 2009; Moulin et al., 2015; Canva, A. et al., 2020). The absence of volcanic evidence and the presence of a deep Moho, points to a relatively thicker crustal block with mafic intrusions (Canva, A. et al., 2020) as the source for anomaly VIII. The RTP intensity values reaches values up to + 70 nT which represents the second highest anomaly values among the study area (Fig.3.A and 7.B). Anomaly VIII outline displays a homogeneous depth around 1-3 km for Euler solutions of SI=0 (Fig.8.A) as previously described by Canva, A. et al. (2020).

### 6.  Conclusions.

The EMAG2v2 magnetic data transformations for the Pyrenees and its foreland basins area have provided new insights into the geological interpretation of the magnetic pattern. The magnetic response for the different domains has portrayed textures that are consistent with the main Pyrenean lineaments. The qualitative interpretation of the magnetic field data has resulted in a comprehensive delineation of eight main distinctive magnetic anomalies: one in the Aquitaine basin (Anomaly VI), three along the North Pyrenean Zone (Anomalies I, II and III), one on the Axial Zone (Anomaly VII), two in the Catalan Coastal Range (Anomalies V and VI) and one on the Gulf of Lyon (Anomaly VIII). Through Euler solutions and power spectrum calculations, depth estimations for each zone has been calculated according to the most relevant morphologies, contacts and sills/dykes. The integration of the geological map and the magnetic transformations has shed light on the intricate pattern of structures and materials that comprise the selected anomalies. Furthermore, the utilization of the tilt derivative method has provided a delineation of the borders of the Axial Zone materials, even though geologically speaking, in the western part the contact between the axial zone and the SPZ is indicated by the presence or absence of the sedimentary cover, while the combination of the horizontal derivatives lineaments has exposed two different structural domains. Including more detailed studies such as petrophysical, structural or geophysical surveys around the areas and anomalies depicted in this work can reduce the uncertainties of the geophysical interpretation provided in this work and it is expected to be part of our future works. Hence, this study is put forward as a groundwork for future

researches aimed to define the magnetic fabric and deep structure of the Pyrenees Mountain Range in specific areas that could be of interest, for instance, to investigate its geothermal potential or mineral resources.

**Data availability**

The data used for this study is available for free in different repositories. Magnetic intensity data has been obtained through EMAG website (https://geomag.colorado.edu/emag2-earth-magnetic-anomaly-grid-2-arc-minute-resolution.html). Despite being available during the development of this paper, EMAG2v2 page seems to be under repair and there is no access available to this data base at the time of the review of this paper. However, the latest version of this data base, EMAG2v3 can be obtained through the link https://www.ncei.noaa.gov/products/earth-magnetic-model-anomaly-grid-. Gravity data for the Pyrenees is available under request and free of charge at IGME (Spanish Geological Survey) repository, from SIGEOF- Geophysical information system (https://info.igme.es/SIGEOF/). Geological and structural maps of the Pyrenees at 1:400.000 can also be obtained for free at IGME repository (http://info.igme.es/cartografiadigital/geologica/mapa.aspx?parent=../tematica/tematicossingulares.aspx&Id=14 &language=es).

**Author contribution**

- Conceptualization: AGM, RS, CA.
- Formal analysis: AGM, RS, CA, TM, FMR, PC.
- Founding acquisition: RS, CA.
- Investigation: AGM, RS, CA, TM, FMR, PC, CRM.
- Methodology: AGM, RS, CA, TM, CRM, JMM.
- Project Administration: RS, CA.
- Software: AGM, CA, TM, CRM, JMM.
- Resources: AGM, CA, JMM.
- Supervision: RS, CA, TM.
- Writing of the original draft: AGM, RS, PC.
- Writing review and editing: AGM, RS, CA, TM, FMR, PC, CRM, JMM.

**Competing interest**

The authors declare that they have no conflict of interest.

**Acknowledgements**

This study is part of the work developed during the "Plan de recuperación, transformación y resiliencia, del Programa Investigo (Convocatoria 2022, Orden de 25 de marzo de 2022, del Consejero de Economía, Hacienda y Empleo, de la Comunidad de Madrid)" and the project PID2020-114273GB-C22 High-resolution imaging of the crustal-scale structure of the Central Pyrenees and role of Variscan inheritance on its geodynamic evolution (IMAGYN) funded by MCIN/AEI/10.13039/501100011033. The authors of this paper are part of CSIC-HUBs (Connections CSIC) - Geosciences for a Sustainable Planet. We also acknowledge the Severo Ochoa extraordinary grants for excellence IGME-CSIC (AECEX2021).

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
