# Peer review of "The geological structures of the Pyrenees and its peripheral basins examined through EMAG2v2 magnetic data."

_EGUsphere, 2024_

## Referee Comment (RC2)

**The structural anatomy of the Pyrenees examined through EMAG2v2 magnetic data.**

África Gamisel-Muzás[1,2], Ruth Soto[2], Conxi Ayala[3], Tania Mochales[2], Félix M. Rubio[2], Pilar Clariana[2], Carmen Rey-Moral[2], Juliana Martín-León[2].

5 [1] Instituto Andaluz de Ciencias de la Tierra, IACT-CSIC, 18100 Armilla, Granada. a.gamisel@csic.es.
[2] Instituto Geológico y Minero de España (IGME), CSIC, 28003, 28760, 50059. r.soto@igme.es, t.mochales@igme.es, fm.rubio@igme.es, p.clariana@igme.es, c.rey@igme.es, j.martin@igme.es.
[3] Geosciences Barcelona (GEO3BCN), CSIC, Lluís Solé i Sabarís s/n, 08028 Barcelona. cayala@geo3bcn.csic.es

10 *Correspondence to:* A. Gamisel-Muzás (a.gamisel@csic.es)

**Abstract.** The major goal of this work is to provide an insight into the structural anatomy of the Pyrenees based on the magnetic data from the Earth Magnetic Anomaly Grid 2-arc-minute resolution (EMAG2v2). We focused on providing qualitative and semi-quantitative evidence on the magnetic signature of the Pyrenees Mountain

15 Range domains and structures. The integration of reduced to the pole and processed maps, as well as the Bouguer anomaly map with geological data, has proved to be significantly useful in order to shed light on the main anomaly sources. Considering their magnetic response and texture, several anomalies can be linked to buried geological bodies or changes in the magnetic character of the basement. We have estimated their source bodies depth through Euler and power spectrum calculations.

20 We have identified eight magnetic zones with different features and interpreted them in terms of the geological and structural setting of the area. The result is an overall interpretation of the Pyrenees main magnetic domains.

**1. Introduction**

[revised manuscript text omitted]

**Fig.4. (A) Vertical derivative map (VDR_Z). (B) Horizontal derivative in X map (HDR_X). (C) Horizontal derivative in Y map (HDR_Y). (D) Horizontal derivative in N110ºE map (HDR_110). (E) Horizontal derivative in N200ºE map (HDR_200). (F) Analytic Signal map (AS). (G) Tilt derivative map (TDR). (H) Horizontal derivative of the tilt derivative map (TDR_THDR). In addition, the main lineaments of each derivative and main structures of the study area are also represented. The main Pyrenean zones are differentiated by diverse texture patterns. MT: Mazamet Thrust, TF: Tolouse Fault, SPT: Sub-Pyrenean Thrust, NPFT: North Pyrenean Frontal Thrust, NPF: North Pyrenean Fault, LTF: La tet Fault, MB: Morreres Back Thrust, LT: Lakora Thrust, SPFT: South Pyrenean Frontal Thrust, VPF: Valles-Penedes Fault. UTM coordinates in m, Zone 31N, ETRS89 datum.**

270

**4.3. Euler deconvolution.**

In order to constrain the depth estimation of the magnetic source bodies, we have used the Euler deconvolution

calculation. As indicated in the methodology, we have chosen the parameters SI=0 (contacts/steps) with a window

of 5 km and SI=1 (sills/dykes) with a window of 10 km. Each window size was selected to ensure the adjusted

275 record of the whole extension of the target bodies.

For the contacts/steps solutions (SI=0, WS=5) (Fig.5.A) the depth estimation reaches up to 5 km to the north of

the North Pyrenean Fault in different clusters, surrounding some of the RTP high positive anomalies, usually near

the outcrops of lherzolites and granulites. The deepest solutions (From 3 km to more than 20 km) are located in

the western sector of the South Pyernean Zone and the Volcanic Field, while shallower solutions (From 0.1 km

280 to 5 km) are found along the Ebro Basin and the Catalan Coastal Range.

[Figure]

For the sills/dykes solutions (SI=1, WS=10), as the window size increases so does the number of solutions (Fig.5.B). Despite showing a similar distribution as in the previous case, their estimations are deeper than in the previous case. North to the North Pyrenean Fault, the solutions display deeper values to the west than to the east, gathered around the higher value RTP anomalies in the zone. Along the Volcanic Field, deep solutions are found in areas without volcanic outcrops. Depth solutions in the western termination of the Ebro Basin, South Pyrenean Zone, Axial Zone and North Pyrenean Zone (From 10 km to more than 20 km) could be related to a deepening of the top of the basement. Values of 10 km are calculated in the border of the Gulf of Lyon offshore RTP great anomaly. On the other hand, the shallowest estimations (0.1 km or less) are found in the center of the RTP anomaly in the Aquitaine Basin, crossed by the Sub-Pyrenean Thrust (Fig.5.B). However, the outcropping materials do not have a relevant magnetic response in the RTP map (Fig.2.B).

[Figure]

[Figure]

what does this mean?

**Fig.5. RTP above 0 nT map in addition to (A) Euler solution for SI:0 WS:5; (B) Euler solution for SI:1 WS:10. Main structures and Pyrenean zones differentiated by diverse texture patterns are also represented. MT: Mazamet Thrust, TF: Tolouse Fault, SPT: Sub-Pyrenean Thrust, NPFT: North Pyrenean Frontal Thrust, NPF: North Pyrenean Fault, LTF: La tet Fault, MB: Morreres Back Thrust, LT: Lakora Thrust, SPFT: South Pyrenean Frontal Thrust, VPF: Valles-Penedes Fault. Volcanic materials, lherzolites, Lagabrielle et al., (2009) lherzolite and granulite mapping, Gulf of Lyon, La Selva, L'Empordà, La Garrotxa and Olot are highlighted. UTM coordinates in m, Zone 31N, ETRS89 datum.**

The purpose of showing anomalies > 0 nT should be clarified.

300    **4.4. Power spectrum.**

[Figure]

which area analyzed, how large a window,
was fractal parameter considered (see Maus and Dimri, 1994
GRL, Ravat et al, 2016, Geosphere)? If fractalness of the field
not considered, then the depths are maximum depth estimates
from the analysis - needs to be mentioned

This specifically discusses Fig. 6 so refer to it first, tell us about why it was chosen.

Maximum depth to the top are around 21-23 km from this plot.

The radially averaged spectrum cosists mainly in three components. A very steep segment (green) that shows a low wavenumber (0.001-0.016 km$^{-1}$). The maximum depth displayed for this segment reaches up to 23 km. The next segment (pink) ranges between wavenumbers from 0.016 to 0.084 km$^{-1}$. This segment displays the shallower sources and noise, reaching up to 15 km. As for the third segment (blue), comprising wavelength values from 0.084 to 0.14 km$^{-1}$ shallower depths are reached, around 12 km. This depth values agree with Andrés et al. (2018) depictions for most of the Curie-Point depths (CDP) in the Pyrenees, pointing to maximum values of 22-24 km depth.

to the top

305

How are the 2D anomaly trends affecting the radially averaged power spectrum interpretation?

Were many such estimates in different parts calculated?

*** This analysis gives the depth to the top of different magnetic layers and not the depth to the bottom. For the depth to the bottom from centroid estimates, see Tanaka et al. (1999, Tectonophysics) or the de-fractal method (Salem et al., 2014, Tectonophysics; Ravat et al., 2016, Geosphere
* * *
[Figure]

[Figure]

This plot shows instantaneous (short wavenumber segment) estimates and it should be labeled and described as such.

This is noise.

**Fig.6. Radially averaged power spectrum of the RTP magnetic field. The wavelength range spans from 7.14 to 125 km. The maximum depth values reach 35 km.**

310

Maximum depth to the top are around 21-23 km from this plot and the slopes of spectra.

**5. Discussion.**

**5.1. Magnetic fabric.**

The magnetic texture within the study area is complex and displays a pronounced heterogeneity in terms of source type, distribution and shape. In order to highlight these parameters through the relationship between the magnetic transformations and the geological structures and lithologies, we have built three merged maps displayed on Fig. 7: (1) A ternary image combining the RTP (Fig.3.A) grid, the Analytic Signal (Fig.4.F) grid and the VDR_Z (Fig.4.A) together with the main lineaments of the HDR_X, HDR_Y, HDR_110 and HDR_200 (Fig.4.B, C, D, E) shown on Fig.7.A; (2) The geological map with the RTP anomalies above 0 nT and the gravimetric maximums (Fig.7.B); (3) The TDR (Fig.4.G) with the main structures and the Axial Zone mapping (Fig.7.C).

315

320

refer to RTP and gravity maps.

to enhance anomalous locations

[Figure]

[Figure]

This is an interesting interpretive map.

Need to say that the derivatives based lineaments are selectively chosen (i.e., not all are there) and the criteria for the choice of features shown.

[Figure]

[Figure]

[Figure]

[Figure]

selectively choosing

as described in the text

[revised manuscript text omitted]

---

## Author Comment (AC1)

[revised manuscript text omitted]
. These materials have been interpreted to be related to mantle exhumation during the crustal hyperextension in the Cretaceous that were later deformed by the Pyrenean shortening (eg. Clerc & Lagabrielle, 2014; Jammes et al., 2009; Lagabrielle et al., 2010; Pedrera et al., 2017). The lherzolite and granulite cartography compilation of Lagabrielle et al. (2010) supports this hypothesis since the outcrops overlain the extension of these anomalies (Fig.7.B).

Euler solutions for SI=0 outline the edge of the anomalies while the clusters of SI=1 suggest depths between 0 and 15 km, commonly between 1 and 5 km and reaching more than 20 km for the northwestern areas of anomaly I (Fig.8.A and 8.B).

In addition, trace of anomalies I, II and III matches exceptionally well with the outline of the gravimetric anomalies (Fig.7.B), also located between the NPFT and the NPF. The shape of these anomalies displayed in the RTP map (Fig.3.A) is a WNW-ESE oriented elongated shape, however the ternary image (Fig.7.A) suggest a more uneven shape that could reveal the source extension at greater depths.

Regarding the Euler solutions, these anomalies are not well defined for SI=0. For SI=1, the deep solutions are found approximately at the edges of the anomalies (up to 15 km) agreeing with Wang et al. (2016) who points to depths between 10 to 30 kilometers for this anomaly source, 
[revised manuscript text omitted]

---

## Author Response (AR1)

Thank you so much for your comments and annotations. We have considered the suggestions you made and marked them in red in the reviewed paper. Here are some comments or explanations (in red), to your notes (in black):

Reviewed text: PYRENEES-EMAG2v2-AGM_30.09_REV1.pdf
Final text: PYRENEES-EMAG2v2-AGM_30.09_REV_TOTAL.pdf

- **Does the paper address relevant scientific questions within the scope of SE?**

This article addresses a dataset extracted from the global magnetic anomaly grid (EMAG2v2) concerning the Pyrenees mountain range and its northern and southern forelands. The scientific question is posed in the title: the study aims to examine the structure of the Pyrenees through magnetic measurements. This study highlights the effectiveness of magnetic anomaly maps in interpreting the structural characteristics of the Pyrenees and its foreland basins. The study is ambitious, as, until now, the geophysical data used in this region (on land) have primarily been gravimetric or seismic.

Okay, thank you.

• **Does the paper present novel concepts, ideas, tools, or data?**

The paper presents a large amount of data in the form of maps, derived from various magnetic data processing methods such as Reduction to Pole (RTP), Magnetic derivatives (VDR_Z, VDR_X, VDR_Y), and also directional derivatives in N110° and N200° adapted to the main Pyrenean structures. Euler deconvolution and power spectrum analysis are also used to constrain the depth of the anomaly sources.
This data constitutes the main contribution of the article, which is purely geophysical in nature. The expected new insights regarding the structures of the Pyrenees and its surroundings are quite disappointing. The interpretations are focused on local areas but do not provide an overall view of the studied zone. As the authors state in their conclusion: 'this study is put forward as a groundwork for future research aimed at defining the magnetic fabric and deep structure of the Pyrenees Mountain Range in specific areas that could be of interest.'

As you mention, the main objective is to provide a groundwork not a detailed or regional scale analysis because the nature of the magnetic anomalies in this area is mainly related with local anomalies. Regarding the new insights, we believe we have stablished a relationship between the Pyrenean structures that are characterized by a magnetic feature, and for the first time, estimated the depth of the causative sources and inferred the origin of some magnetic anomalies that are not cropping out.

• **Are substantial conclusions reached?**

The substantive conclusions focus on the identification of 8 major magnetic anomalies, for which the authors have estimated the depths of the source bodies by making assumptions about their geological nature. But these interpretations are not sufficiently

discussed or defended in light of the current geological knowledge of the Pyrenees and its surroundings.

Thank you, we tried to add more information as comparation to previous geological works.

- **Are the scientific methods and assumptions valid and clearly outlined?**

The geophysical processing methods are clearly described and seemingly executed flawlessly. However, the question of the structural interpretation of these various data in geological terms is unclear. We do not really know what the authors' intention is regarding the geological interpretation of their magnetic data.

Thank you, our main objective is to interpret the magnetic anomalies in relation to the causative bodies. The ones that crop out are already known but we have estimated their depth. Also, we have hypothesized the geological origin of magnetic anomalies in places where there are no outcrops of "magnetic bodies" and evaluated its approximated depth. Moreover, the tilt derivative has provided a better delineation of the Axial zone. Finally, our study is a groundwork for more detailed investigations.

- **Do they want to compare the geophysical data with recent global models of the mountain range, or do they simply intend to offer their processed magnetic data results to the community without delving further into structural interpretation?**

In any case, the article lacks an explanation of how the magnetic response of rocks relates to their lithology. This may seem obvious to the authors, but not all readers are specialists in this field.
The Pyrenees are a well-known mountain range today, and there is a consensus on its overall structure.
In my opinion, the goal of the article should be to identify the structures that are highlighted by the magnetic anomalies as well as those that are not. Based on these observations, an attempt should be made to provide an explanation.

As to how the magnetic response correlates to the lithologies, several references are mentioned in the paper. As this work is not a methodological study, we see no need to further the explanations on how the methods work.
As we have indicated in your previous query, we have identified the structures that are highlighted by the magnetic anomalies, the ones that crop out and the ones that do not crop out but generated positive anomalies, which is one of the main contributions of the paper. We cannot evaluate the structures that do not have a magnetic response. In this way, we have worked from the magnetic response and analyze what can be seen and interpreted from it. So, we decided to focus on the main positive magnetic anomalies since they play a fundamental role in the deep structures understanding.

- **Are the results sufficient to support the interpretations and conclusions?**

The results presented in the form of maps derived from various geophysical processing methods are very consistent and represent a remarkable data source with significant study potential for the entire geoscience community.

However, the geological interpretations provided do not live up to the quality of the geophysical work accomplished.

We have tried to complete the geological interpretation.

Section 5.2.

- **Is the description of experiments and calculations sufficiently complete and precise to allow their reproduction by fellow scientists (traceability of results)?**

Nowhere is there mention of the formulas used for the various magnetic data processing methods."

As this is not a methodological work, we think it is not necessary to add the formulae to the description of the methods used in the paper. We have provided the relevant references where the reader can find the corresponding formulae.

- **Does the title clearly reflect the contents of the paper?**

The title should be completed e.g. "The structures of the Pyrenees and its peripheral basins examined through EMAG2v2 magnetic data.". The term 'anatomy' seems too ambitious as it suggests a more detailed description of the structure, the shape of objects, and the relationships between them, which
the paper does not provide.

Thank you, we agree with your comment and changed the terminology used as you suggested.

- **Does the abstract provide a concise and complete summary?**

The abstract accurately reflects the methodology but provides little information about the study's results. The last sentence: "The result is an overall interpretation of the main Pyrenes domain" does not really convey the results of the work that focuses only on certain positive anomalies but does not address the entire area.

Thank you. The last sentence of the abstract already talks about the identification of the eight most relevant magnetic anomalies, which are the focus of the paper.

L.20 in the reviewed text. L. 22-26 in the final text.

- **Is the overall presentation well-structured and clear?**

The presentation of the results is well-structured:

Chapter 1 Introduction, Chapter 2 Methodology, Chapter 3 Geological Setting, Chapter 4 Data Processing and Analysis (the largest section), Chapter 5 Discussion.

It shows a clear progression in the approach and clearly distinguishes between data an interpretation. The language is fluent and clear. However, Chapter 3 is somewhat brief. Chapter 5 should be titled 'Interpretation of the Data.' A substantial discussion chapter on the contribution of magnetic data to structural geology is missing. It should also be noted that there are no references in the text to the calculation methods (formulas) used in the processing, unlike in many geophysics articles. An opinion from a geophysicist on this point is essential.

Thank you. We respectfully disagree on the "there are no references in the text to the calculation methods" since each method on the "Data and methodology" section has the most relevant references that contain the formulae of the corresponding method.
We agree that we should add a discussion on the contribution of the magnetic data to structural geology, even though that is not the main objective of the paper.

• **Are mathematical formulae, symbols, abbreviations, and units correctly defined and used?**

Surprisingly, there are no calculation formulas in the magnetic data processing, which seems quite unexpected in this type of paper. Not being a geophysicist, I cannot answer with certainty about the geophysical units used, which seem standard to me. However, it seems strange that the Tilt Derivative (TDR) or TDR_THDR data maps are labeled with units (nT/m), because this type of transformation should be unitless.

Should any parts of the paper (text, formulae, figures, tables) be clarified, reduced, combined, or eliminated?

All the requested or suggested modifications are detailed in the specific comments.

As mentioned before, this is not a methodological paper, so we believe that, in order to keep the paper to the point, which is the geological interpretation of the magnetic anomalies, no formulas are needed since this information is in the references and the references therein.
The units of Tilt and the other derivatives units are the same as used in many other papers.

• **Are the number and quality of references appropriate?**

The bibliography is quite comprehensive, and it is surprising that, given the richness of the references, the geological section is so brief. A few missing references are suggested in the comments.

The new references are considered. The reason of the briefness in the geological section is in fact the richness of the bibliography. Since this is not a geological depiction paper, if the reader needs deeper knowledge about the geology of the study area, we provide this abundant bibliography.

• **Conclusion**

In summary, this paper does not seem mature enough to be published in its current form. The geological context is not sufficiently well presented at the beginning of the article regarding the essential structural features of the study area.

An uninformed reader will not be able to understand the interpretations provided by the authors. In my opinion, current geological cross-sections are essential to make the article more accessible to everyone.

Thank you for your comments. We hope that the new additions and modifications provide a better understanding of the point of the study and its interpretations.

Additionally, the interpretations are presented on a case-by-case basis without proper context. The geological map (actually lithological) that is frequently mentioned is not very readable and too detailed in respect of the magnetic anomaly maps.

The detail on the lithological map is necessary since the outcrops of some rocks are regional-scale but also critical to the interpretation of the anomalies.

Specifically for Fig 3A, the foundation of the article, the legend colors used are very confusing for a non-expert. It would be more intuitive to set the color yellow at 0 nT. However, such a representation may not be traditional in geophysics...

The color scale is provided by the software and since it does not hinder the objective of the map and its read is intuitive, we did not consider its modification.

It is not entirely clear what the difference is between Chapter 5.1 (Magnetic Fabric) and 5.2 (Geological Interpretation). Both already provide geological interpretations. The interpretations are generally not sufficiently explained and justified, and they are often guided by existing outcrops without taking into account the numerous models that are widely accepted by the geological community.

The difference between the two sections is that 5.1 provides a depiction of the magnetic response according to the area features, while the 5.2 goes through the main zones and each anomaly sources within them.

Finally, the conclusion lacks an assessment of what the magnetic data reveal in relation to the current knowledge of the Pyrenees and its key areas. The Labourd anomaly, which is the subject of numerous articles, is mentioned in just three lines. What about the North Pyrenean massifs? Are the thrust sheets of the Axial Zone or the Sierra Marginales visible through the magnetic anomalies? More generally, could the Pyrenean chain be detected solely through magnetic data? (It appears to be invisible in the eastern third of the range...).

Thank you, we took it into account and we have tried to be more specific about the descriptions but the main objective Is to describe the units that can be detected through

magnetic data not depict the whole Pyrenees features and then focus on the ones that can be registered trough magnetic data.

**Specific comments**

- **Abstract:**

If the goal is to focus on the identification of 8 positive anomalies, these should be highlighted at the beginning of the article.

Thank you, this is highlighted now.

L.20 in the reviewed text. L. 22-26 in the final text.

- **2. Data and Methodology**

The paragraph is very clear and informative for non geophysicists.

A few suggestions:

L70: It might be helpful to provide more details on the formula used for the calculation (to be confirmed by a geophysicist).

We believe that the references given in the text are enough should a reader be interested in the formulae. Since this is not a methodological work, we feel there is no need to add the formulae.

L91: Explain what is meant by "depth" for an anomalous source (is it the top contact of the source or a center of gravity?).

The depth of the anomalous sources refers to the center of the magnetic source, whose approximated shape is given, in the case of Euler Solutions, by the structural index.

L96: For the structural index, what is meant by "contact" or "steps" (for S0), and "dike" or "sills" (forS1)? There is also other SI (spherical shape ? ) — why was it not used?

As the seequent guide reads: A structural index is an exponential factor corresponding to the rate at which the field falls off with distance, for a source of a given geometry.
This way, 4 SI can be used: 0-contact/step, 1-sill/dike, 2-cylinder/pipe, 3-sphere/barrel/ordnance. We tried all of them for our data, but considering the believed morphology of the source bodies and the inconsistent results for SI:2 and SI:3, we opted for the use of SI:0 and SI:1.

- **3. Geological Setting of the Study Area**

L124: Cite Wher et al., 2018.

Okay.

L126: The Pyrenean structural zones are presented in maps (Fig. 2), but it would also be necessary to show them in crosssections (e.g., ECORS profiles, Beaumont 2000, or more recent crosssections from Ford et al. 2022, as cited by the authors).

Noted.

See Figure 1.

L133: Also cite Cochelin et al. 2017.

Okay.

L135: (Insert) Permian to Cretaceous, the latter being associated with the opening of the North Atlantic…

Okay.

L137: Dolerite, known as ophite, derived.

Okay.

L144- 145 Fig 2 A&B: The two maps are overly complex and detailed compared to the magnetic data (resolution 3.7 km). These maps are rarely used in the subsequent analysis. Were all the lithologies presented in Figure 2B actually considered in this study? It would be better to use a map that maintain the structures (faults and thrusts) already indicated and highlights the main lithologies based on their supposed magnetic susceptibilities. For example, for the basement: distinguish granites/gneisses, pelitic metasediments, and if possible, carbonates (Devonian). For the cover: preCretaceous carbonate sediments, Cretaceous and Eocene terrigenous sediments. Units such as the marginal sierras should also be included on these maps.

The detail in the maps is necessary for the magnetic analysis.

L156: These authors did not define the 5 Pyrenean zones.

We added some more references that will be helpful for the readers interested on the definition of the Pyrenean Zones: Mattauer, M. (1968). Les traits structuraux essentiels

de la chaîne Pyrénéenne. Revue de Géographie Physique et Géologie Dynamique, 10(1), 3–11.

L.161. in the reviewed text. L167 in the final text.

L166 : Introduce a cross-section from Ford et al. 2022 to illustrate the Aquitaine Basin. Similarly, all other zones need to be shown in crosssection (NPZ, AZ, etc.) (L166 to L189).

Noted.

L173: Also mention the North Pyrenean massifs that might be visible on the magnetic maps.

Since we describe only the visible features we see no need to name them.

L180: Also cite Saspiturry et al. 2017.

Okay, I believe you mean 2019?

L.183 in the reviewed text. L.192 in the final text.

**4. Data Analysis and Results**

L194: It is necessary here to specify the relationship between lithology and magnetism based on the existing rock types in the study area, as presented in Figure 2B (which was suggested to be simplified earlier in L145). It is also important to provide the limitations of interpreting anomaly maps.

Okay, we added some explanation.

L.199-208 in the reviewed text. L. 206-215 in the final text.

For instance, the potential discrepancy between the structural geological map, which only shows surface structures, and the magnetic anomaly map. Similarly, the magnetic response can vary depending on whether the contacts are vertical or gently inclined. Essentially, the goal is to provide interpretative guidelines for a geologist who may not be familiar with magnetic anomaly maps.

L196: Why cite Pedreira here? A few lines comparing the magnetic anomaly map and the Bouguer anomaly map would be welcome here.

The one cited here is Pedrera not Pedreira.

**Figures 3 A-B-C and subsequent figures:**

For better localization, it would be necessary to include on these maps (and subsequent ones) the structures from Figure 2, which are much more comprehensive (e.g., including

the Gavarnie thrust, Sierra Marginales, etc.). It would also be useful to include some localities such as Figueres, Balaguer, Zaragoza, etc. The use of hatched or cross-hatched zones on the maps is not particularly useful. Readers can easily navigate without these graphic overloads.

We considered this option but the intersection between lines and colors made the maps unreadable. As for the localities, we agree on this and added some on the first maps.

The Bouguer anomaly maps are not clearly visible in Figures 3B & C. It would be necessary to show a standalone Bouguer anomaly map in Figure 3B to allow for comparison with Figure 3A.

Noted.

See figure 3.

L210-211: Frankly, at first glance at Figures 3, it is difficult to see a clear correspondence between the anomaly map and the structural zones of the Pyrenees (except for the western part of the NPZ where 3 anomalies are aligned and also correspond to gravity anomalies). What can be observed in Figure 3A is a negative anomaly domain covering the western and central parts of the Pyrenees and the Aquitaine Basin, intersected by alignments of positive anomalies corresponding to the North Pyrenean Zone. In the Axial Zone, the negative anomaly seems to correspond to the area where the crustal prism is thickest and shows at the surface an increasingly complete Paleozoic sedimentary sequence, while to the east, where the anomalies are positive, the crust becomes progressively thinner towards the Mediterranean Sea, and it is in this area that the ages of the metasediments are the oldest (pre-Devonian).

This is discussed later on in the paper.

L216: These anomalies could be related to the overall shape of the Pyrenean crustal prism (Chevrot et al., 2018). While the presence of thick evaporite layers can be suggested to explain the positive anomalies in the Sub-Pyrenean Zone, the same explanation cannot be applied to the negative anomaly in the Axial Zone, which does not contain evaporites.

Noted.

L223 : What do the authors mean by 'also originated by buried volcanic rocks'? This volcanism is very recent—are they referring to thicker-than-expected volcanic deposits or a subsurface feeder system beneath the surface volcanism?

We point to a subsurface feeder system beneath the surface that cannot be inferred by the outcrops alone.

L.233 in the reviewed text. L. 238 in the final text.

L224 : Stating that 'the relative maxima are related to outcrops of Paleozoic basement' is not sufficient, as the Paleozoic also outcrops throughout the entire Axial Zone. It may be necessary to compare this zone with the Ebro Basin, which contains a significant thickness of Neogene sediments.

Noted.

L225 : A summary of the 8 main anomalies, which are the key focus of the paper according to the authors, should be provided here.

As the anomalies that we identify are the result of the interpretation of the magnetic maps, we consider that is more logical to present them once we have already shown and explained the magnetic maps.

**4.2 Magnetic derivatives**

Fig 4 ABCDEFGH : As mentioned earlier, the structural overlay should be more complete (use the one from Fig. 2 and add a few localities). Also remove the hatched patterns.

If we use the complete structural mapping and the localities we hinder the reading of the maps. That is why we use the line patterns to define the different tectonic zones.

Not being a geophysicist, I wonder why the TDR map (Fig. 4G) and the TDR_THDR map (Fig. 4H) have units in nT/m?

The units of the TDR and TDR_THDR are nT/m or radians, we choose the first option. See reference Verduzco et al., 2004.

Additionally, I don't understand why the HDR_X (N90°) and HDR_110 images do not look almost similar with only a 20° difference? The same goes for HDR_Y (N180°) and HDR_200. Could there be a mix-up in the figures.

The "small" difference between the angles is enough to highlight different structures therefore generate significantly distinct maps.

There is not mix-up in the figures, we have double-checked every result and every figure.

Finally, I am unsure how to interpret the differences in the anomaly values in the legend, which use the same color ranges (e.g., -0.009 for HDR_X&Y (Fig. 4B&C) and -0.99 for HDR_110&200). In my view, it would be necessary to explain, in the text, these differences in magnitude if we want to compare the various structures from the different derivative maps.

As you see in the color bars, the scales are really different and that is because each of them depends on different values (dip, angles, etc), the main objective of these derivative maps is to be used in a qualitative way not in a quantitative way. The derrivatives highlight the lineaments due to changes within a given lithology, litholigcal

contacts, faults and other variations in magnetic character, the color scales are only to highlight the maxima and minima of the derivatives.

L229 : to the North of the southerm border of the Axial Zone

Noted.

L.239 in the reviewed text. L.245 in the final text.

232 to 240 : Is it really necessary to show all four horizontal derivative maps? Wouldn't it be better to keep only HDR_110 and HDR_200? It seems that the structures outlined in white on HDR_X&Y are even better represented on HDR_110&200.

Traditionally, HDR_X and HDR_Y are always shown since they are the first ones to be calculated. In our case, it provides us a better image of the magnetic response and the lineaments and is also useful as a comparative to the HDR_110 and HDR_200 that highlight the main Pyrenean directions.

L240 : HDR_200 (Fig. 4E) does not seem to be sufficiently utilized. It shows a diffraction of structures that transition from a NNW-SSE orientation in the southern part of the range to a WNW-SSE orientation in the western and central parts, and then back to a NNW-SSE direction in the Aquitaine Basin. The eastern part of the Pyrenees, however, maintains a NNW-SSE (Variscan?) direction. But I am not sure that this Fig. 4E actually represents HDR_200 because it is the N-S lineaments that are highlighted.

As you mentioned before, the 20º difference is not that big but is indeed significative. In the case of the HDR_Y the N200ºE structures are also identified but as you can see, the definition of their traces is not as clear. The HDR_200 may also register N-S structures, but the definition is clearly better than for N200ºE lineaments

L241 : The phrase 'the NW-SE half' can be confusing because it does not specify which part is being referred to.

Noted.

L.251 in the reviewed text. L257 in the final text.

L245 to 250 : In terms of structural orientation, the TDR map (Fig. 4G) does not fundamentally differ from the derivative map (Fig. 4E).

It may look like they are the same but there are not, and the features they highlight are different. Both are necessary.

**4.3 Euler deconvolution.**

L271-272 : It would be better to explain, for a non-expert reader, the meaning of Sills/Dykes (S1) and Contacts/Steps (S0). Are these purely geological definitions, or do they hold specific significance for geophysicists? Indeed, a spatial resolution of 3.7 km cannot reveal dykes (in the geological sense) in the Pyrenean context.

Moreover, the geometry of a sill and a dyke are very different, as are contacts and steps. It is also necessary to clarify what is meant by the depth of the source body—does it refer to the contact point between the source body and its surroundings, a sort of center of gravity, or something else?

They define structures, and as I mentioned before they are just referencing morphologies in order to apply the Euler deconvolution in the most faithful way.

Also better explain in the text that the points are calculated on the contacts of the RTP map. Similarly, clarify (if it is the case) that Euler deconvolution typically assumes a single, dominant source contact for each solution, rather than a combination of different objects. It is not clear how the Euler points are calculated.

Is it a choice made by the authors? Why does the density of points differ? What do the alignments of points correspond to? All of this deserves an explanation for non-geophysicists.

This is just how the method works, as the paper is not a methodological paper, the non-geophysycist readers can use the references in order to learn about the method implementation. The density and location of the points depends on the structural index and the window chosen for the calculation.

Fig. 5 : Is there a way to simplify/summarize these raw point maps, particularly in Figure 8 ?

We are sorry, it is not possible to simplify the points since each of them is a result of the calculation and each solution is crucial.

**4.4 Power spectrum :**

L301 to 307 : There is an ambiguity between 'maximum depth' for the first segment and 'shallower depth' for the other two segments. Additionally, the significance of this radially averaged spectrum calculation should be better explained. What is the relationship with the depths previously obtained using Euler deconvolution? How can the two approaches be reconciled? The radially averaged power does not have a map representation like the previous depth measurements.

Noted, we have tried to clarify the sentences. Both methods are approaches to the estimated depths for the anomalies but none of them is can be interpreted as a precise depth but as an approximation. The Euler Solutions give the depths over an area (hence the representation in a map) whereas the Power Spectrum shows the estimated depth according to the wavelength of the anomaly and it is represented in a graph so you can compare which depth is expected for a given wavelength.

See section 4.4.

**5 – Discussion.**

In my opinion, this chapter should be titled 'Structural Interpretation.' The focus should be on demonstrating how magnetic anomalies can reveal the structure of the Pyrenees (based on currently accepted models) and also on identifying which recognized structures are not visible through the magnetic data.

Noted.

See section 5.

Fig. 7 A : The colored AS-dZ-RTP triangle (top right) is not explained.

That is the color scale, we pointed it in the figure description.

See Figure 7 explanatory text.

Fig. 7B : Lithological map too detailed (same as Fig. 2B).

As mentioned before, it is necessary to support our interpretations.

Fig.7C : It should be overlaid with the structures from Fig. 2A

It would hinder the readability of the map.

L348 : « E-W structures linked to the compressive structure of the Pyrenees… » This statement cannot be disputed, but it should not be forgotten that the axes of Variscan deformation in the Pyrenees are the same as the axes of the Cretaceous-Tertiary orogeny.

Okay.

L349 : The interpretation of N-S structures as being related to the opening of the NW Mediterranean Basin is only valid along the coast and, of course, does not apply to the other structures inland.

Noted.

L351 : « related to the evaporites to the south-west » is not precise enough. Are we referring to the southwest of the study area, in the Ebro Basin? It would be necessary to clarify why the evaporites would result in N-S structures. Note that these N-S structures are not indicated on Fig. 4B (HDR_X).

Okay.

L353 to 357 : This sentence about the volcanic structure needs to be clarified.

Okay.

L358 : This lineament on the Catalan coast also appears in the Bouguer anomalies (likely related to crustal thinning). Attributing this magnetic anomaly to an intrusion is just one hypothesis among others, and it's too general. The context of this intrusion should be specified.

Noted.

L358 -360 : It would be appropriate here to better explain how evaporitic structures would generate this signal. Are we talking about diapirs? What is the age of the evaporites? Are they characteristic of the region? The authors write as if everyone is familiar with the geology of the area.

Okay. References in the geological interpretation.

L360-362 : This statement is not really illustrated in the figure. It would be better to indicate on which of the different maps the Tet Fault appears.

It is illustrated since two lines, one at each side of La Tet Fault , are mapped in the referenced map (7.A) (Pink lines).

L364 : Constrain the configuration of the Axial Zone in its western and central parts, where the thickness of the Pyrenean crustal prism is greatest. Pathway of the North Pyrenean Fault , except in its the East part.

Okay.

L365 to 370 : It may seem strange that the Lakora Thrust is marked by a magnetic lineament, as this thrust is gently dipping to the north and buried beneath the North Pyrenean Zone... In fact, I believe there is confusion regarding the representation of the Lakora Thrust, which corresponding fact to the southward thrusting of the southern edge of the NPZ. The authors seem to extend this thrust south of the Axial Zone, which causes confusion in relation to the terminology of the Pyrenees.

We are sorry, we have just noticed the mistake in the thrust name, we refer to Larra thrust not Lakora.

L370-373 : A N200E lineament…..La tet Fault : These lineaments in question are not found on Figures 7A & B. This sentence is quite unclear in relation to Figure 7.

Figure 7.A shows the lineaments that you refer, they are the pink lines that appear to both sides (west and east) from La Tet Fault.

**5.2 Geological interpretation of the magnetic map.**

The paragraph 5.1 is already partially a structural interpretation…
In fact, in this chapter, the intention is only to focus on the positive anomalies. It would be better, in this case, to title it 'Geological Interpretation of the Positive Magnetic Anomalies.

Noted.

See section 5.2.

Fig. 8 A&B : Is it possible to simplify these point maps?
Additionally, one or two cross-sections, for example from Ford et al. 2022, would help in understanding the structures of this chapter. These have already been suggested for section 3.1.

No, it is not possible to simplify the point maps since each of them is a result.

**5.2.1 Aquitaine Basin**

L388 : Ensure that Le Maire et al. (and not Marie) are indeed referring to this anomaly and not the one in Saint Gaudens (idem for Pedrera et al. 2017 L 390)

Okay, we fixed the mistake in the reference.

L.404 in the reviewed text. L.410 in the final text.

L391 to 393 : The SPT is indicated as 'inferred' in Figure 2 and subsequent figures. It corresponds to  very shallow and gently dipping thrust zone. Therefore, a 'late feature' cannot be inferred. The 'cut' of this anomaly is probably only apparent.

We are sorry there was a misunderstanding, we refer to a coincidence in the location not a structural cut. We have changed the terminology.
Also, the inferring of this feature was previously provided by other authors (added now in the text):

Angrand, P., Ford, M., & Watts, A. B. (2018). Lateral variations in foreland flexure of a rifted continental margin: The Aquitaine Basin (SW France). Tectonics, 37. https://doi.org/10.1002/2017TC004670

Rougier, G., Ford, M., Christophoul, F., & Bader, A.-G. (2016). Stratigraphic and tectonic studies in the central Aquitaine Basin, northern Pyrenees: Constraints on the subsidence and deformation history of a retro-foreland basin. Comptes Rendus Geoscience, 348(3–4), 224–235. https://doi.org/10.1016/j.crte.2015.12.005

L.417-421 in the reviewed text. L.417-420 in the final text.

Moreover, considering a dyke of this size (30 km in diameter) associated with the SPT is likely the most improbable hypothesis in the Pyrenean context. The simplest hypothesis would be to consider Anomaly IV in the same way as Anomaly III, interpreted as slivers of more or less serpentinized mantle (see Espurt et al. 2019 in Tectonophysics). or as a huge cretaceous sills (Teschenite) in the sediments.

Both hypotheses are considered, but we added of references to our discussion so the main points of the are based and understood.

L.417-421 in the reviewed text. L.417-420 in the final text.

L396 : Again, are we certain that Chevrot et al. (2018) are indeed referring to this anomaly and not the one in Saint Gaudens (Anomaly III)? These authors have never mentioned dykes for these anomalies. What exactly do the authors mean by a dyke structure (SI=1)?

Yes, we have checked what anomaly they talk about and these authors refer as Saint Gaudens (Anomaly III) as the southern anomaly, not Saint Gaudens II (Anomaly IV) but some mistakes may have occurred in some other references, making the definition of these two anomalies confusing.
A dyke-like structure just implies the possibility of a vertical structure and not necessarily a dyke stricto-senso.

L399 : anomaly IV and not VI

Noted. There was a mistake in the sentence.

398 to 400 : The sentence is not clear at all.

Noted. There was a mistake in the sentence.

Finally, this chapter lacks an explanation for the most negative anomalies, which are located in the Aquitaine Basin, even though the authors prefer to focus on positive anomalies.
Should the presence of thick evaporite layers in a deep basin be considered?

The main objective of the paper is the identification and interpretation of the main anomalies which are positive.

**5.2.2 North Pyrenean zone**

It may be necessary to remind in this part, that this area is now explained in terms of crustal hyperextension, which led to mantle exhumation that was later reactivate during Pyrenean shortening.

Noted.

See section 5.2.2.

L402 : granulites are not mantle material

Noted.

L404 : Even though lherzolite bodies outcrop, they are relatively small in size compared to the spatial resolution of the anomalies and are therefore not directly responsible for the anomalies. They support the hypothesis of larger bodies at depth. In the eastern Pyrenees (south of Saint Barthelemy or even at Lherz), mantle outcrops do not produce magnetic anomalies despite their extent.

The fact that some outcrops are small in size in comparison to the area of the magnetic anomaly they generate is because the magnetic method not only considers the surface lithologies but also their extension in the subsurface. The reason why the mantle materials you name do not produce a magnetic response could be due to different reasons, one of them being that the rocks were formed at temperatures higher than the Curie point and therefore demagnetized.

L413 to 421 : Here, the authors must absolutely reference and compare their estimated depths for the Labourd anomaly with Wang et al. 2016 (GSA and not Geology)...

Okay

L416 : « Probably related to Molasse Neogene » is unclear and requires an explanation.

Thank you for the comment, we have just realized we made a mistake in the anomaly description and no Neogene Molasse is related to the anomaly depth or environment, we meant the IV anomaly and instead of related we meant beneath.

**5.2.3 Axial zone**

L426 : As with the other studied areas, an overall geological cross-section would be useful to illustrate the global structure of the Axial Zone (e.g., cross-sections in Ford et al. 2022).

Okay.

See figure 1.

L427 - 428 : This assertion about the low magnetic susceptibility of granitoids seems quite surprising given their highly variable content of ferromagnetic minerals.

The granitoids referred on the text are the ones at the Pyrenees that have been extensively studied in terms of its magnetic response (Porquet et al., 2017). A comprehensive study of the magnetic character (or lack of) in the Pyrenean granites can be found in Pueyo et al., 2022 and references therein.

L428 : I suggest stating that the anomalies decreasing at the eastern and western ends of the Axial Zone could reflect the thinning of the crustal prism. The WNW-ESE magnetic lineaments characterizing the area are probably related to the elongated metamorphic domes resulting from Variscan deformation (Cochelin et al. 2017), which were reactivated by thick-skinned tectonics during the Cretaceous-Tertiary orogeny.

We can not infer that from our data, it would be necessary to add more detailed studies.

L431-433 : The authors need to justify the existence of Cambrian meta-basalts and Pleistocene volcanism here, with references. This Anomaly VII (which is also described, but differently, L 468) extends along the entire coast and cuts across all the Pyrenean zones, so why not invoke mantle exhumation due to the opening of the Gulf of Lion? C

Noted, references were added.

L.476 in the reviewed text. L.451 in the final text.

L439 : The term 'buried' is ambiguous because it can suggest either tectonic burial or burial under a thick sedimentary deposit, which is impossible given the Pleistocene age of the volcanism. Do the authors mean that the volcanism is much more extensive at the surface and at depth than is currently known?

Yes, we imply that the extension of the volcanic field is bigger underneath the surface that what could seem by the outcrops.

L.482 in the reviewed text. L.456 in the final text.

**5.2.4 South Pyrenean zone**

L442-442 : Given its depth, it is possible that the source bodies of this anomaly (up to 6 nT) are located in the footwall of the SPFT ?

We have clarified this question in the original text, by adding this paragraph: Magnetic anomaly sources within the area located at depths below 2-6 km (SI=1) might be linked to sources located in the footwall of the South Pyrenean Thrust according with the geometry interpreted of Pyrenean crustal cross sections (see Teixell et al., 2018).

L.496 in the reviewed text. L.469 in the final text.

L442 to 445 : the authors need to clarify why this value is explained by the presence of evaporites and ophite. Do the estimated depths correspond to the interface between the basement and the cover?

The defined depth is related to the presence of shallow ophites within the evaporites. We have modified the text to clarify this question: "In this area, small and dull responsive magnetic anomalies have been linked to the presence of underlying Triassic ophites by Zeyen and Banda (1989)."

L.492 in the reviewed text. L.465 in the final text.

**5.2.5 Catalan costal Range**

L454 - 456 : This assumption of attributing the positive Anomaly VI to the presence of granitoids is contradictory to the statement in lines 427-428, which says that granitoids have a very low magnetic susceptibility. Why not also invoke the presence of granitoids with ferromagnetic minerals for Anomaly VIII in that case?

Thank you for pointing out our mistake, we realized our contradiction and modified the text in order to explain the anomaly sources for anomalies V and VI due to basalts.

L.510 in the reviewed text. L. 481 in the final text.

**5.2.6 Ebro Basin**

L462 : Il would be more accurate to say : The RTP map displays anomaly values above 0 nT on its western and eastern edges, and exceptionally in the center, extending from those previously described to the south of the SPZ.

Okay.

L.517 in the reviewed text. L. 489 in the final text.

L462 -463 : Again, the hypothesis of attributing these positive anomalies to the Triassic is not the best solution in my opinion. Geological cross-sections (e.g., Ford et al. 2022 and many others) show that the SPFT is emergent here and at shallow depth (<2 km). The continuity between the anomalies of the two zones (SPZ & Ebro Basin) can only be considered if these anomalies are located in the basement, below the SPFT. If that were not the case, then the SPFT would cut through the anomaly and would not cause any displacement.

We do not have enough data to confirm this theory but we are interested in exploring this question.

**5.2.7 Gulf of Lyon.**

L468 to 471 : This explanation is convincing but contradictory to lines L431 to 433, which suggest that Anomaly VII is sourced from Cambrian metabasalts and Pleistocene volcanic rocks.

We clarified the source origin for this anomaly, with concordance to lines 431 to 433.

L.525 in the reviewed text. L.496 in the final text.

**6. Conclusions.**

L482 : It is better to refer here to the Pyrenees and its foreland basins.

Okay.

L.539 in the reviewed text. L.510 in the final text.

L485 : You need to specify 8 main distinctive positive anomalies.

Noted.

L.543 in the reviewed text. L.513 in the final text.

L489-490 : In reality, the AZ and SPZ are defined at the surface only by the contact between the basement and the cover, which is not always steeply dip. Thus, the difference between the Axial Zone and the South Pyrenean Zone, at the contact (in the west part) is only indicated by the presence or absence of Mesozoic or Cenozoic cover. This type of information should be mentioned at the beginning of the article.

Okay, we pointed out your remark.

L.549 in the reviewed text. L.520 in the final text.

L491-492 : Regarding the study area, the authors cannot speak of an 'absence of more detailed studies such as petrophysical, structural, or geophysical surveys.' In recent years, major Franco-Spanish projects have produced a considerable amount of data, which the authors may not have fully utilized, even though some are cited in their bibliography (this is one of the critiques of this paper).

Okay, thank you, we agree. The sentence can be misinterpreted.

L.551-553 in the reviewed text. L.522-524 in the final text.

Final remark on Chapter 5: Chapter 5, which should be titled 'Structural Interpretation of the Magnetic Data,' addresses the depth of source bodies solely based on Euler deconvolution calculations.

Okay.

See section 5.

What about the 'radially averaged spectrum,' which is never mentioned again in the text?

The uncertainty of its results may give us an approximate depth but can not be fully trusted for the main interpretations.

Thank you so much for your comments and annotations. We have considered the suggestions you made and marked them in green in the reviewed paper. Here are some comments or explanations (in green), to your notes (in black):

Reviewed text: PYRENEES-EMAG2v2-AGM_30.09_REV1+REV2.pdf
Final text: PYRENEES-EMAG2v2-AGM_30.09_REV_TOTAL.pdf

L.42. Find out and describe survey characteristics of datasets that went into this compilation and because any limitations of the grid spacing, line spacing, survey elevation, draping or not will affect the anomaly artifacts and ultimate resolution of this dataset and its interpretation.

- As stated in the text, the survey elevation is 4000 m ant the spatial resolution is 2-arc-min; more details about the surveys that make up EMAG2v2 can be found in Maus et al., 2009.
L.55 in the reviewed text. L.55 in the final text.

L.57. This long-winded description of the methods is distracting. Other than the names of the methods and key references, move the descriptions to where you are first describing the results and showing the maps.

- As we have been discussing with the other referee, who requested an extension of this description, we believe that pointing the main objective of each method and some references should be enough for non-geophysics. However, we believe that these brief explanations must remain in the paper since the audience of this paper is not only geophysics-expertise scientists, as the other referee pointed out.

L.190. Is the inducing field assumption valid here? Aren't there a number of sources here whose remanent magnetization may be significant? Provide explanation.

- As remnant magnetization data indicates a low remanence and we do not have data about the remnant magnetization vector, we chose the RTP mapping since it does not take into account the remnant magnetization effect (we have made the analysis in rock samples and we have a paper under review where this data will be published).

L.210. Actually there are some anomalies of greater wavelengths than this in the SW part of the map. What is this wavelength range relevant for? Is this wavelength range not only for the window later used for spectral analysis?

- This value of the wavelength is only informative, the process of calculating the racially averaged power spectrum gives the wavenumber content of the magnetic signal, we have converted wavenumber to wavelength because we think this will be clearer for the reader, in general.

-L.305. which area analyzed, how large a window, was fractal parameter considered (see Maus and Dimri, 1994 GRL, Ravat et al, 2016, Geosphere)? If fractalness of the field not considered, then the depths are maximum depth estimates from the analysis - needs to be mentioned.

Sorry, we have not considered any fractal parameter.

How are the 2D anomaly trends affecting the radially averaged power spectrum interpretation?

We have calculated the averaged power spectrum directly from the grid. Therefore, we have not separated the effect of 2D anomaly trends.

Were many such estimates in different parts calculated?

No, we have made the calculation for the whole area.

*** This analysis gives the depth to the top of different magnetic layers and not the depth to the bottom. For the depth to the bottom from centroid estimates, see Tanaka et al. (1999,Tectonophysics) or the de-fractal method (Salem et al., 2014, Tectonophysics; Ravat et al., 2016, Geosphere )

Yes, thank you for the observation. We have modified the text accordingly.

L.322 in the reviewed text. L. 317 in the final text.

This plot shows instantaneous (short wavenumber segment) estimates and it should be labeled and described as such

Sorry, we do not understand the question. The data correspond to the moment of the data acquisition

L.394. why does the dyke make circular feature? Even at this great depth it couldn't be a long dyke? Why not just call it an intrusion?

When we talk about dike we just refer to a morphology that could generate a symmetrical anomaly as the one we see here. Since it has little lateral variation we infer a vertical dike like body.

L.415. Given how the Euler solutions are calculated based on gradients in a window, the interpretation of both SI=0 and 1 solutions as a edges of source bodies at shallow and deep parts seems difficult to digest.

We used the Euler Solutions to estimate if the magnetic sources are deeper or shallower than the neighboring anomaly sources, we are aware that the depths we have obtained have a great uncertainty.

L.453. seems likely that granites would intrude through basaltic volcanic formations instead of underlying the granitic bodies. Is there any other evidence for buried volcanic material here? If yes, give references that show this relationship.

Thank you for the comment. Following the other referee and your comment, we have already corrected the text.

L.519 in the reviewed text. L.478 in the final text.

+ Attending to the link of the data base, the National Centers of Environmental Information (NCEI) from the National Oceanic and Atmospheric Administration (NOAA) has been reached and no answer arrived. It seems that the page is under maintenance but it was not the case at the time this study was in progress. Any news about the availability of the data will be provided. However, the latest version of this data base is available:

-EMAG2v3: https://www.ncei.noaa.gov/products/earth-magnetic-model-anomaly-grid-2

---

## Referee Report (RR1)

The work presented here provides a significant contribution in terms of the quantity of geophysical data offered. The numerous maps effectively illustrate this contribution, even for non-expert readers. The introduction is ok.

The following section then focuses on the presentation of the data. A key missing element here is an explanation of how the maps were produced, as well as the different numerical and mathematical methods used to process the geophysical data. There are no tables or illustrations to show the analysis of anomalies at depth, no calculation tables, and no discussion comparing the data used to other published works, such as the cited publications by Zeyen et al. (1988, 1989, 1991).

The presentation of the geological context correctly sets the scene and provides a good overall understanding, albeit slight, of the objective. However, the structural aspect is rather weak given the objective of the study. A better presentation of the geological framework would help to understand what the geological finalities are.

As it stands, there is no real discussion, and the hypotheses put forward rely on uncertain comparisons with interpretations from other publications, without providing the reader with comparative figures (structural cross-sections, cross-analysis of data, etc.). Thus, even without going into detail on the methodology and results, the "interpretation" section alone demonstrates that this work is not an in-depth analysis of the Pyrenean structure but rather a simple comparative catalog of correlated positive anomalies. There is no discussion of the geological nature of the data used or of the geological structures that the authors consider most plausible.

In its current state, while the work is rich in ideas and data, it is incomplete—both in terms of data analysis (including the presentation of the methodology, which lacks critical perspective) and in the study of geological structures and the presentation of existing structural studies (cross-sections and maps) that could support the demonstration.

A critical analysis component is still lacking, likely because the scope of the study is too broad, preventing a detailed analysis of each zone and its associated anomalies.

Positioned between two disciplines—the presentation of geophysical data and structural analysis—the current version of the manuscript does not fully belong to either.

Finally, a comparison of the previous reviewers' reports with the implemented corrections shows that key considerations regarding the manuscript's maturity and necessary improvements have not been addressed. The minor corrections made fail to tackle the essential points that have already been highlighted in previous reviews.

Given the revision work already undertaken, I am compelled to reject the current version and request that the authors undertake a thorough revision of their article.

[revised manuscript text omitted]

---

## Author Response (AR3)

Editor comment
Response
Original version
Corrected version.

Dear Authors,
I accept your responses to the referee.
Some requests are not addressed, although I agree that the information is most of the time accessible via the literature.
One exception concerns the Oasis Montaj software, for which you should provide some references.
best regards

Thank you for your comment.

The Oasis Montaj Software incorporates the formulae described in the references provided for each method in the corresponding section:

-RTP: Baranov, 1957; Baranov and Naudy, 1964.

-Magnetic derivatives: Verduzco et al., 2004; Fanton et al., 2014; Hayatudeen et al., 2021; Ansari and Alamdar, 2009; Miller and Singh, 1994; Verduzco et al., 2004; Blakely et al., 2016; Lahti and Tuomo, 2010; Ma and Li, 2012.

- Euler deconvolution: Thomson, 1982; Reid et al., 1990; Cooper, 2008; FitzGerald et al., 2004.

- Power spectrum: Maus and Dimri, 1995; Peredo et al., 2021; Spector and Grant, 1970.

Global methods for magnetic interpretation, specifically reduction to the magnetic pole (RTP), Magnetic Derivatives and Analytic Signal are described in detail in Blakely, 1995, as for the equations for the Euler solutions, they are described on Thomson, 1982, and for the Power Spectrum the methodology is described in Spector and Grant, 1970.

We also added a reference to show the procedures and principles followed by Oasis Montaj for each transformation: Hinze et al., 2013.

Original:

To enhance the interpretation of the magnetic signal of the Pyrenees, derivative maps and estimated calculations of the location and depth of the magnetic sources have been generated from the total magnetic field. This enables us to improve and complete the correlation between geophysical and geological information. These calculations were made through the Oasis Montaj© software by Seequent.

Corrected version:

To enhance the interpretation of the magnetic signal of the Pyrenees, derivative maps and estimated calculations of the location and depth of the magnetic sources have been generated from the total magnetic field. This enables us to improve and complete the correlation between geophysical and geological information. These calculations were made through the Oasis Montaj© software by Seequent (Hinze et al., 2013) whose formulae is primarily based on the works of Blakely (1995), Thomson (1982) and Spector and Grant (1970) and further developed considering the references included in the description of each methodology.